# Targeting SOX10-deficient cells to reduce the dormant-invasive phenotype state in melanoma

Claudia Capparelli[1,2✉], Timothy J. Purwin [1], McKenna Glasheen [1], Signe Caksa [1], Manoela Tiago[1], Nicole Wilski [1], Danielle Pomante[1], Sheera Rosenbaum[1], Mai Q. Nguyen[1], Weijia Cai [1], Janusz Franco-Barraza[3], Richard Zheng[4], Gaurav Kumar[1,2], Inna Chervoneva [5,6], Ayako Shimada [5,6], Vito W. Rebecca[7,8], Adam E. Snook [2,6], Kim Hookim[9], Xiaowei Xu[10], Edna Cukierman [3], Meenhard Herlyn [7] & Andrew E. Aplin [1,2✉]

Cellular plasticity contributes to intra-tumoral heterogeneity and phenotype switching, which enable adaptation to metastatic microenvironments and resistance to therapies. Mechanisms underlying tumor cell plasticity remain poorly understood. SOX10, a neural crest lineage transcription factor, is heterogeneously expressed in melanomas. Loss of SOX10 reduces proliferation, leads to invasive properties, including the expression of mesenchymal genes and extracellular matrix, and promotes tolerance to BRAF and/or MEK inhibitors. We identify the class of cellular inhibitor of apoptosis protein-1/2 (cIAP1/2) inhibitors as inducing cell death selectively in SOX10-deficient cells. Targeted therapy selects for SOX10 knockout cells underscoring their drug tolerant properties. Combining cIAP1/2 inhibitor with BRAF/MEK inhibitors delays the onset of acquired resistance in melanomas in vivo. These data suggest that SOX10 mediates phenotypic switching in cutaneous melanoma to produce a targeted inhibitor tolerant state that is likely a prelude to the acquisition of resistance. Furthermore, we provide a therapeutic strategy to selectively eliminate SOX10-deficient cells.

[1] Department of Cancer Biology, Thomas Jefferson University, Philadelphia, PA 19107, USA. [2] Sidney Kimmel Cancer Center, Thomas Jefferson University, Philadelphia, PA 19107, USA. [3] Cancer Signaling and Epigenetics Program, Marvin & Concetta Greenberg Pancreatic Cancer Institute, Fox Chase Cancer Center, Philadelphia, PA 19111, USA. [4] Department of Surgery, Thomas Jefferson University, Philadelphia, PA 19107, USA. [5] Division of Biostatistics, Thomas Jefferson University, Philadelphia, PA 19107, USA. [6] Department of Pharmacology & Experimental Therapeutics, Thomas Jefferson University, Philadelphia, PA 19107, USA. [7] Melanoma Research Center, The Wistar Institute, Philadelphia, PA 19104, USA. [8] Biochemistry and Molecular Biology Department, Johns Hopkins University Bloomberg School of Public Health, Baltimore, MD 21205, USA. [9] Department of Pathology, Anatomy and Cell Biology, Thomas Jefferson University, Philadelphia, PA 19107, USA. [10] Department of Pathology and Laboratory Medicine, Perelman School of Medicine, University of Pennsylvania, Philadelphia, PA 19104, USA. ✉email: Claudia.Capparelli@Jefferson.edu; Andrew.Aplin@Jefferson.edu

ntratumoral heterogeneity and cellular plasticity between states with differing proliferative, invasive, and drug phenotypes allow tumors to adapt to foreign microenvironments during the metastatic process[1]. While it is known that cellular plasticity involves reprogramming of transcriptional profiles, the mechanisms that mediate phenotypic switching are poorly characterized. Heterogeneity and plasticity are evident in cutaneous melanoma and are associated with non-responsiveness and acquired resistance to pharmacological inhibitors of the BRAF-MEK signaling pathway and to immune checkpoint antibodies such as anti-PD1 and anti-CTLA4[2–4].

Melanomas arise from the transformation of neural crest (NC)-derived cells. These highly aggressive tumors frequently show remarkable growth inhibition when treated with targeted BRAF inhibitors (BRAFi) and/or MEK inhibitors (MEKi). However, these effects are transient, and eventually acquired resistance ensues. Early studies identified differential expression levels of microphthalmia-associated transcription factor (MITF) in melanoma associated with distinct characteristic states and transcriptomic signatures[5]. The proliferative state was associated with high levels of MITF, whereas the invasive state exhibited low MITF but high expression of the epithelial-to-mesenchymal transcription factor, ZEB1, and TGFβ signaling genes[6]. Differential expression of MITF, SRY (sex-determining region Y)-box 10 (SOX10), and BRN2/POU3F2 transcription factors, or p75 NGFR, define the invasive and proliferative states and provide further subclassification[2,7–13]. Despite this knowledge, the factors required for phenotypic switching in melanoma remain poorly defined.

SOX10 is highly expressed during early NC development and maintained in the migratory phase but is repressed as NC cells differentiate. SOX10 expression is used as a marker for melanoma and its role is intriguing. High SOX10 expression positively regulates melanoma cell proliferation, tumor growth, and invasion[14,15]. In addition, SOX10 has been linked to response to targeted therapies. A SOX10-deficient, slow-cycling state has been implicated in acquired resistance of BRAF-mutant melanoma to BRAFi through upregulation of TGFβ1-EGFR signaling[16]. In other studies, elevated MITF expression and an MITF-low/AXL-high state have both been associated with BRAFi resistance[17,18].

Stable acquired resistance may arise from an innate or adaptive drug-tolerant state. Recent single-cell sequencing studies have characterized proliferative/melanocytic, undifferentiated/invasive, and neural crest-like drug-tolerant states, highlighting a non-mutational phenotypic adaptation[2,9,11,12,19]. Here, we show that SOX10 is heterogeneously expressed in melanomas independently of treatment status. SOX10 loss is sufficient to induce an invasive slow-cycling state and tolerance to BRAFi and/or MEKi, which together likely form the basis for acquired resistance. Importantly, we identify a vulnerability of SOX10-deficient melanoma cells based on the up-regulation of cellular inhibitors of apoptosis-2 (cIAP2). Together, our findings provide a strategy to target SOX10-deficient subpopulations and reduce drug-tolerant populations in melanoma.

## Results

**SOX10 expression is heterogeneous in melanoma.** To identify mediators of phenotype switching, we assessed SOX10 expression at the single-cell level in melanoma patient samples using a publicly available single-cell RNA-sequencing (scRNA-seq) dataset[20] and observed both inter and intratumoral heterogeneity (Fig. 1a, b). Thirteen of the 14 tumors analyzed were BRAF inhibitor treatment-naïve. In addition, seven tumors were immune checkpoint inhibitor (ICi) treatment-naïve (TN) and seven were resistant to immune checkpoint inhibitors (ICR). Of

note, ICi-resistant patient tumors Mel110 and Mel106 were homogeneously negative for SOX10 expression (Fig. 1a, b). In the other 12 melanomas, SOX10-expressing and SOX10-deficient cells co-existed within the same tumor lesion. Evidence of intratumoral heterogeneity was supported further at the protein level by IHC analysis of melanomas showing clusters of tumor cells with either low or high expression of SOX10 (Fig. 1c). To further validate SOX10 heterogeneity in melanoma, we analyzed a tumor microarray (TMA) obtained from 62 melanoma patient samples[21] (Fig. 1d). Twenty-eight samples (45%) were treatment-naïve, 8 samples (13%) were treated with targeted therapy, 18 samples (29%) with ICi, and 7 samples (11%) with targeted therapy and ICi. One of the tumor samples lacked treatment information. Four samples were completely negative for SOX10, while all the other tumor samples presented high intertumoral and intratumoral-heterogeneity, corroborating our observations from the scRNA-seq dataset[20]. SOX10 immunofluorescence analysis in A375 (*BRAF* V600E), MeWo (wild-type for *BRAF* and *NRAS*), and 1205Lu (*BRAF* V600E) melanoma cell lines showed a similar level of heterogeneity and confirmed the co-presence of SOX10-low and -high expressing cells in basal culture conditions (Fig. 1e-g). Of note, we observed fewer SOX10-deficient cells in MeWo compared to the A375 and 1205Lu cell lines (data not shown). Together, these data show that SOX10 is heterogeneously expressed in melanoma and SOX10-low populations are present in treatment-naïve samples.

**SOX10 loss causes transcriptomic changes associated with an invasive phenotype.** Given the presence of SOX10-negative cells in pretreatment melanoma, we sought to characterize this subpopulation. We used CRISPR/Cas9 and multiple guides to knock out SOX10 in MeWo and A375 cells and generated multiple clones with loss of SOX10 expression for each cell line (Fig. 2a). RNA-seq analysis and Gene Set Enrichment Analysis (GSEA) were performed to characterize the SOX10-regulated transcriptome and evaluate significant pathway changes following SOX10 knockout (Fig. 2b, c and Supplementary Fig. 1A, B). In SOX10 knockout cells, we observed an enrichment in pathways associated with the tumor microenvironment (epithelial-mesenchymal transition, TGFβ signaling, extracellular structure organization, apical junction, hypoxia, and angiogenesis) and metabolism (glycolysis pathway), as well as enrichment of the p53 pathway and TNFA signaling via NFκB (Fig. 2b, c and Supplementary Fig. 1A, B). Additionally, we observed reductions in MYC and E2F targets (Supplementary Fig. 1A). We further corroborated the involvement of SOX10 in regulating these pathways using two publicly available datasets[11,16] containing SOX10 knockdown transcriptome data from A375 cells and 6 different patient-derived melanoma cell lines (MM001, MM011, MM031, MM057, MM074, MM087) (Supplementary Fig. 1C–E).

To determine the role of SOX10 in the transition between phenotypic states, we compared CRISPR knockout SOX10 cells with parental cell lines using the proliferative and invasive signatures identified by Verfaillie, et al.[6]. SOX10 knockout cell lines showed enrichment of invasive signature genes and a reduction in proliferative signature genes (Fig. 2d and Supplementary Fig. 1F, G). Similar results were observed when the same analysis was performed in datasets from SOX10 knockdown melanoma cell lines[11,16] (Supplementary Fig. 1H). These data show that SOX10 loss is sufficient to lead to transcriptional alterations associated with phenotype switching to a low proliferative/high invasive state.

**SOX10 depletion induces an invasive-like state.** To further characterize the proteomic alterations associated with differential

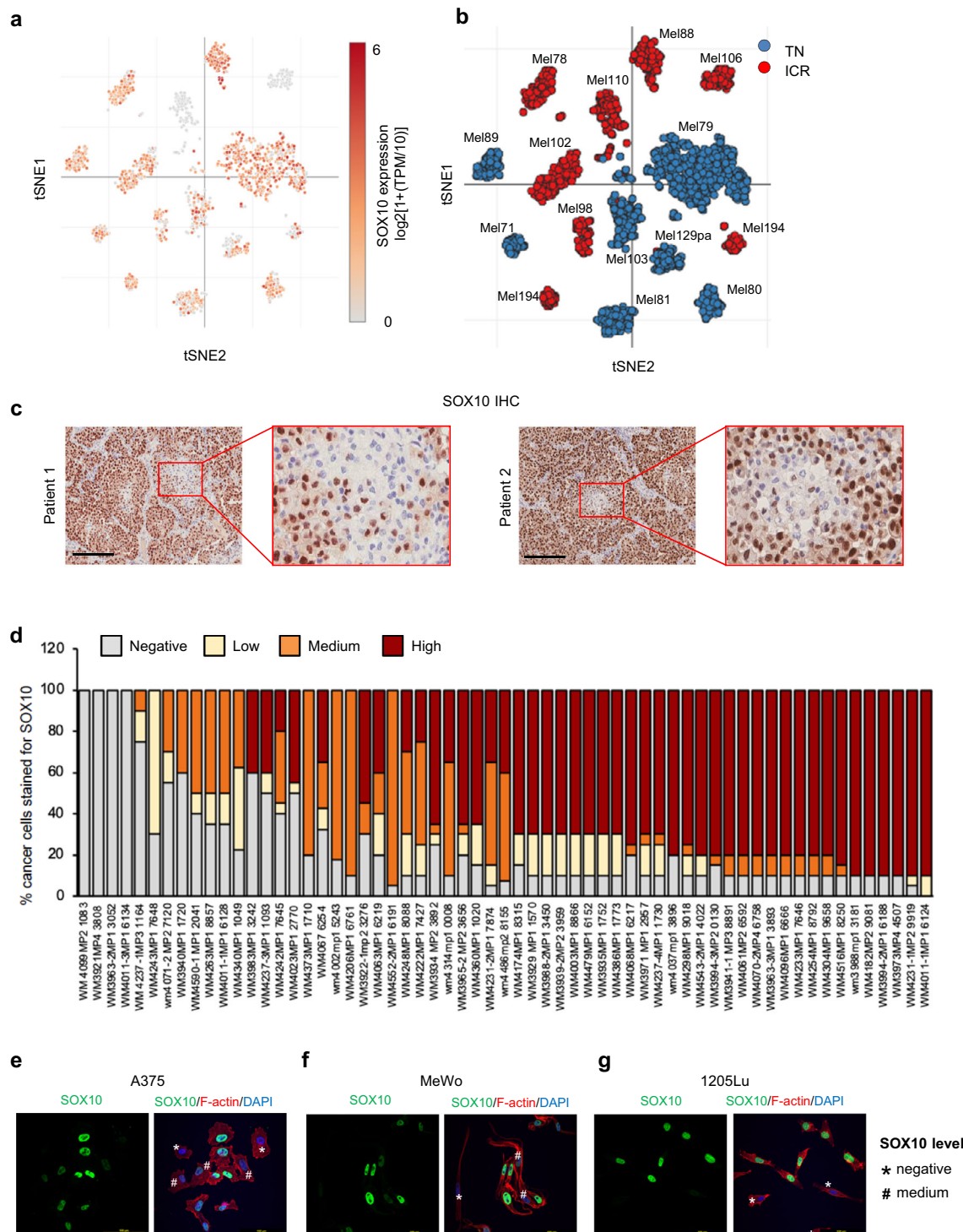

**Fig. 1 Melanoma patient samples show heterogeneous SOX10 expression. a** tSNE plot of scRNA-seq cells colored by SOX10 expression level from Jerby-Arnon, et al. **b** As in **a** except cells colored by treatment. **c** Melanoma patient samples from Thomas Jefferson Hospital stained for SOX10. Scale bars, 200 μm. **d** TMA samples stained for SOX10. In each sample, the percentage of cancer cells expressing none, low, medium, or high levels of SOX10 was quantified by a pathologist (Dr. Xu). Samples were sorted based on their H-score (low to high). **e** Immunofluorescence images of A375, cells stained for SOX10 (green), filamentous actin (red), and DAPI (blue). The experiment was performed independently twice, and representative images are shown. Scale bar, 100 μm. **f** As in **e** except MeWo cells were used. **g** As in **e** except 1205Lu cells were used.

expression of SOX10, we analyzed SOX10 knockout cells by western blotting. We first confirmed that, as previously suggested, SOX10 regulates the expression of ErbB3[22] and PDGFRβ[16]. MeWo SOX10 knockout cells displayed increased protein levels of the invasive markers fibronectin (FN1), ZEB1, WNT5, N-cadherin, and tyrosine-phosphorylated focal adhesion kinase (FAK) (Fig. 3a). In addition, we detected reduced levels of the proliferative markers MITF, phospho-RB1, and cyclin D3, as well as upregulation in the cyclin-dependent kinase inhibitor, p21[Cip1] (Fig. 3a). To address the concern of inter-clonal heterogeneity, we

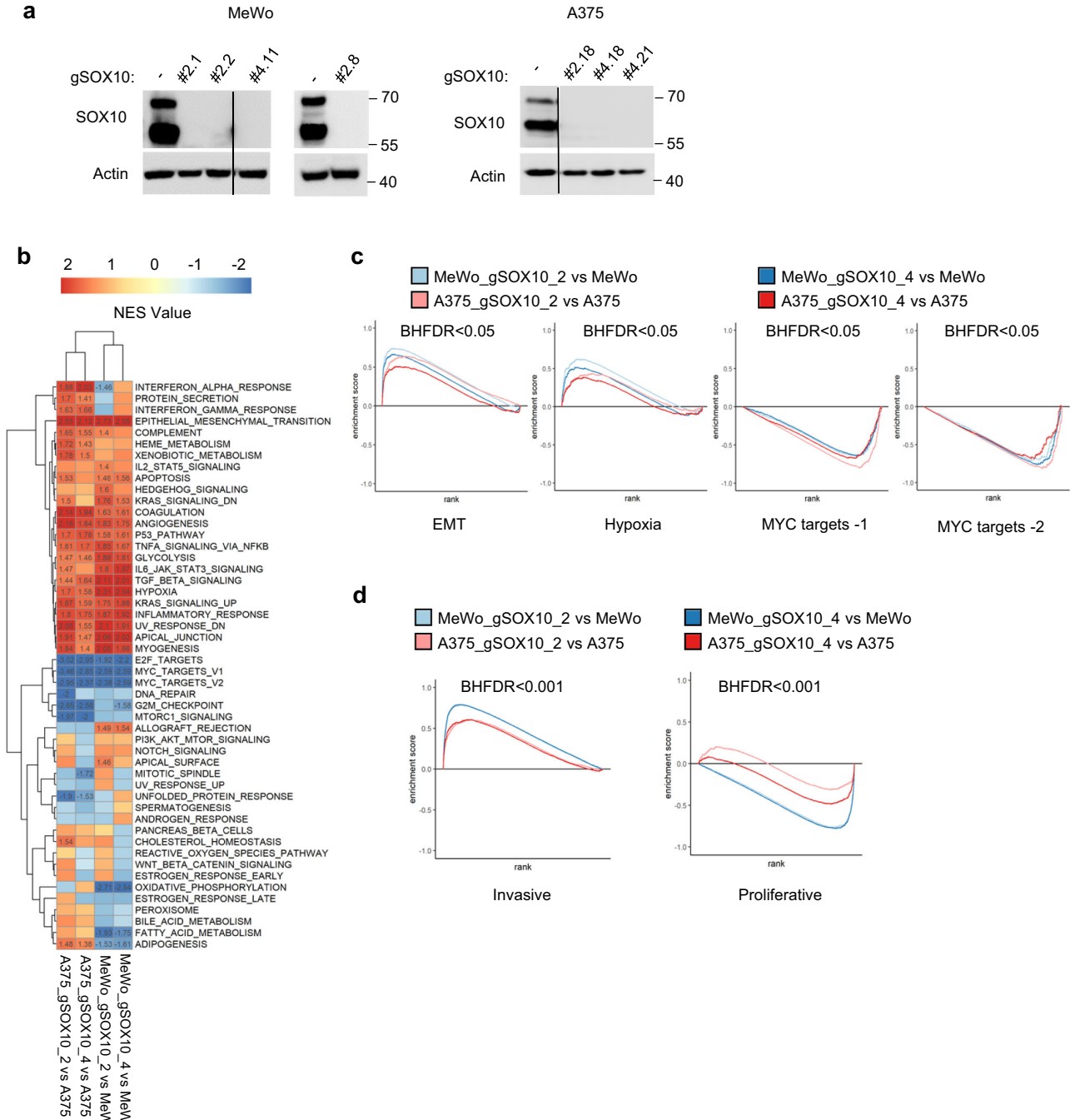

**Fig. 2 SOX10 regulates the transcription of genes involved in invasion, proliferation, and cell metabolism pathways. a** MeWo and A375 SOX10 knockout cell lines were generated as described in Materials and Methods. The same number of parental and SOX10 knockout cells were seeded in six-well plates for each cell line. Cells were lysed and lysates western blotted as indicated. The experiment was repeated independently three times with similar results. **b** A heatmap showing GSEA normalized enrichment scores (NES) for the hallmark gene sets collection for MeWo and A375 SOX10 knockout cells (guide #2 and #4) compared to parental cells. NES values are displayed for enriched gene sets, using a Benjamini-Hochberg False Discovery Rate (BHFDR) cutoff of 0.05. MeWo gRNA#2 includes combined data collected from clones # 2.1, # 2.2, and # 2.8. For all other samples (MeWo gRNA #4, and A375 gRNA#2 and gRNA#4), shown is the mean from three independent replicates generated for each clone. **c** Enrichment plots of EMT, Hypoxia, MYC targets-1 and MYC targets-2 comparing MeWo and A375 SOX10 knockout cells (guide #2 and #4) with parental cells. **d** Enrichment plots of proliferative and invasive gene signature (Verfaille, et al.) for MeWo and A375 CRISPR SOX10 knockout (guide #2 and #4) vs parental cells. BHFDR < 0.001.

additionally co-mixed the four MeWo SOX10 knockout clones at equal ratios to generate a SOX10 knockout pool. Consistently, we observed similar changes in invasive and proliferative markers in the MeWo SOX10 knockout pool and A375 SOX10 knockout cells when compared to MeWo and A375 parental cells, respectively (Supplementary Fig. 2A, B).

Based on the upregulation of fibronectin and increased tyrosine phosphorylation of FAK, we further explored alterations in extracellular matrix (ECM) production and remodeling. FN1 is often secreted by the mesenchymal compartment whereas collagen IV is commonly produced by tumor cells. Consistent with the mesenchymal phenotype observed in SOX10-deficient

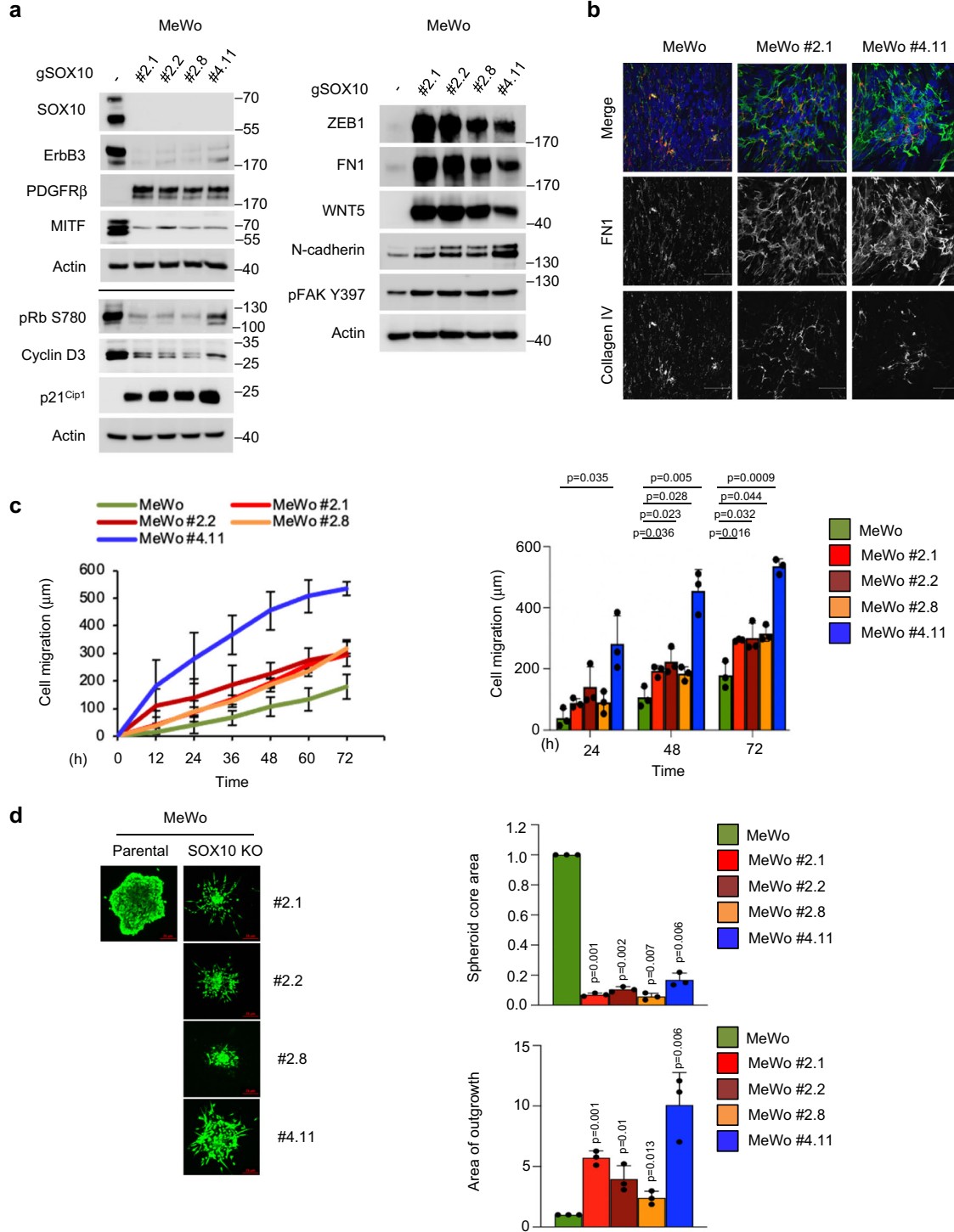

**Fig. 3 SOX10 regulates the expression of genes associated with a slow-cycling and more invasive phenotype. a** The same number of MeWo parental and MeWo CRISPR SOX10 knockout (clone # 2.1; 2.2; 2.8; 4.11) cells were seeded in six-well plates for each cell line. Cells were lysed and lysates western blotted as indicated. The experiment was repeated independently two times with similar results. **b** MeWo parental, MeWo #2.1, and MeWo #4.11 cells were plated on coverslips coated with 0.2% gelatin. The next day, cells were treated with ascorbic acid (50 μg/ml) for 6 days. Treatment was renewed every 48 h. At the end of the experiment, cells were permeabilized, fixed, and stained for FN1 and collagen IV. The experiment was performed independently twice, and representative images are shown. Scale bars, 50 μm. **c** Scratch-wound assay comparing MeWo CRISPR SOX10 knockout (clones # 2.1; 2.2; 2.8; 4.11) cells to parental cells. Shown is the mean ± SD from three independent experiments. *p*-values were calculated using a two-sided *t*-test and *p*-values for significant comparisons are shown. **d** Spheroid in 3D collagen comparing MeWo CRISPR SOX10 knockout (clones # 2.1; 2.2; 2.8; 4.11) cell lines to parental cells. Shown is the mean ± SD from three independent experiments. Scale bar, 25 μm. *p*-values were calculated using a two-sided one-sample *t*-test of the null hypothesis and *p*-values for each comparison are shown.

cells, extracellular deposition of FN1 appeared increasingly organized in fibrillar structures in SOX10 knockout MeWo cells compared to the parental cells (Fig. 3b and Supplementary Fig. 2C, D). Conversely, no change was observed in collagen IV (Fig. 3b). These results suggest that SOX10 alters ECM production. Since cancer cell-derived ECM is associated with invasion[23], we performed migration and invasion assays. In 2D IncuCyte scratch-wound healing assays, SOX10 knockout cells traveled significantly farther than controls (Fig. 3c and Supplementary Fig. 2E). In 3D collagen spheroid assays, SOX10 knockouts displayed decreased core area but increased outgrowth (Fig. 3d and Supplementary Fig. 2F). These data suggest that SOX10 loss induces an invasive state associated with increased production of ECM components organized in fibrillar structures.

**SOX10 loss is sufficient to induce a quiescent/dormant-like phenotype in vivo.** Since cancer cell-derived ECM is associated with invasion[23] and dormancy[24], to determine if the enhanced invasive properties of SOX10 knockout cells were associated with a slow-cycling phenotype, we examined the growth of luciferase-expressing MeWo intradermal xenografts. Parental MeWo cells generated tumors that typically reached a palpable size within two weeks (Fig. 4a). In contrast, SOX10 knockout MeWo cells failed to form palpable tumors, but luciferase imaging showed the presence of residual cancer cells at the sites of injection (Fig. 4b and Supplementary Fig. 3A). These residual cells remained detectable at three months post-injection (Fig. 4c). IHC staining of residual tumors (at day 35) confirmed the lack of SOX10 expression (Supplementary Fig. 3B) and showed significantly reduced Ki67 and increased p21$^{Cip1}$ levels compared to control tumors, indicating reduced proliferation (Fig. 4d and Supplementary Fig. 3C). Next, to evaluate whether SOX10 knockout causes alterations in the ECM in vivo, we used second-harmonic generation of polarized light microscopy to visualize polymerized/fibrous collagen bundles at the residual tumor areas. Consistent with the in vitro data (Fig. 3b), SOX10 knockout tumors showed enrichment in fibrous ECM signatures (Fig. 4e, f and Supplementary Fig. 3D). Overall, our in vivo studies suggest that SOX10 knockout cells display a quiescent/dormant-like phenotype associated with increased fibrous ECM protein deposition/organization.

**SOX10 is downregulated in melanomas resistant to MAPK targeting agents.** Loss of SOX10 has been previously linked to acquired resistance to BRAFi in *BRAF*-mutant melanoma[16]. These findings prompted us to characterize SOX10 deficiency in the context of targeted therapy tolerance. First, we compared the effect of MAPK targeting agents in parental versus SOX10 knockout cells in both *BRAF* wild-type and *BRAF*-mutant models. Parental MeWo and A375 cells were highly sensitive to either MEKi or BRAFi + MEKi treatments, respectively (Fig. 5a and Supplementary Fig. 4A-C). By contrast, SOX10 knockout cells were significantly less sensitive to MEKi and BRAFi + MEKi treatments (Fig. 5a and Supplementary Fig. 4A-C). In the A375 parental cells, we detected reduced SOX10 expression in the surviving population following 14 days of BRAFi + MEKi treatment (Supplementary Fig. 4D). To investigate whether SOX10-deficient cells preferentially survive MEKi, we co-mixed optically bar-coded mCherry-SOX10-proficient and GFP-SOX10-deficient cells at a 2:1 ratio. We used this ratio since SOX10-negative/low cells often appear to be a subclonal population in melanoma patient samples. In the absence of treatment, SOX10-proficient cells grew out, whereas chronic exposure to MEKi selected for SOX10-deficient cells (Fig. 5b and Supplementary Fig. 4E).

One possibility is that acquired resistance to targeted therapy arises, at least in part, from cells in a drug-tolerant state. We analyzed the expression of SOX10 in cell lines previously generated in vitro and in vivo that are tolerant/resistant to BRAFi or BRAFi + MEKi[1,25]. All tolerant/resistant cell lines analyzed showed dramatically reduced levels of SOX10 expression compared to their parental counterparts (Fig. 5c, d and Supplementary Fig. 4F). The expression of SOX10 was not affected by the presence/absence of the inhibitors, suggesting that the loss of SOX10 is mediated by neither acute drug administration[26–28] nor by addiction to the inhibitors that can develop following acquired resistance[29,30]. Of note, SOX10 loss was more evident in cell lines generated in vivo compared to in vitro. Consistent with SOX10 knockout cells (Fig. 2c, d and Supplementary Fig. 1A–H), GSEA analysis of tolerant/resistant lines, namely, A375 CRTs (CRT34 and CRT35, combination BRAFi/MEKi-resistant tumors) and 1205Lu PBRTs (PBRT15 and PBRT16 BRAFi-resistant tumors), showed enrichment in epithelial-mesenchymal transition (EMT), TGFβ signaling, apical junction, hypoxia, glycolysis, and TNFα signaling via NFκB, as well as a reduction in MYC and E2F targets (Supplementary Fig. 4G–I). Furthermore, the resistant/tolerant cells (CRTs and PBRTs) displayed an enriched invasive gene signature and reduced proliferative gene signature[6] compared to their respective parental cells (A375 and 1205Lu) (Supplementary Fig. 4J, K). When comparing the transcriptomic profiles of SOX10-deficient cells that arose following BRAFi + MEKi treatment in vivo (CRT35 and CRT34) to SOX10 CRISPR knockout cells generated in vitro, we observed a similar transcriptomic pathway enrichment and gene regulation (Fig. 5e, f). These data suggest that many of the pathway alterations observed in the CRT lines may be driven by the repression of SOX10.

Consistent with the SOX10 knockout phenotype, CRT34 and CRT35 showed an increase in ECM protein deposition (Fig. 5g, in the presence of BRAFi/MEKi, and Supplementary Fig. 4L, at basal condition) and increased outgrowth in 3D collagen in comparison to A375 parental cells (Supplementary Fig. 4M). Next, we queried published RNA-seq datasets from matching patient samples before and after acquiring resistance to MAPK targeting agents[16]. Consistent with the cell line-derived RNA-seq data (Fig. 2c, d and Supplementary Fig. 1A–H and 4G–I), GSEA analysis showed that resistant tumors lacking SOX10 or with a SOX10-deficient transcriptomic profile displayed enrichment in genes involved in EMT, hypoxia, TGFβ signaling, apical junction, angiogenesis, TNFA signaling via NFκB, glycolysis, and extracellular structure organization (Fig. 5h and Supplementary Fig. 4N). Resistant tumors also displayed an enriched invasive gene signature and a reduced proliferative gene signature[6] compared to pretreatment biopsies (Fig. 5i). In the transcriptomic profile of CRT lines, many pathways associated with proliferation (E2F targets and MYC targets) were downregulated in the presence of BRAFi + MEKi, consistent with drug-tolerant (versus fully resistant) properties (Supplementary Fig. 4O). Consistent with reduced G1/S progression, RB1 phosphorylation was decreased and p27$^{Kip1}$ expression was increased following BRAFi + MEKi treatment of CRT lines (Supplementary Fig. 4P). These data indicate that a drug-tolerant SOX10-negative population often arises following the treatment of melanoma with BRAFi and/or MEKi, and that this might be the population of cells that seeds tumor recurrence.

**SOX10-negative cells are sensitive to cIAP inhibitors.** Our data indicate that SOX10 knockout cells display a targeted therapy-tolerant state associated with invasive properties. Next, we sought to identify drugs that are selectively lethal for SOX10 knockout

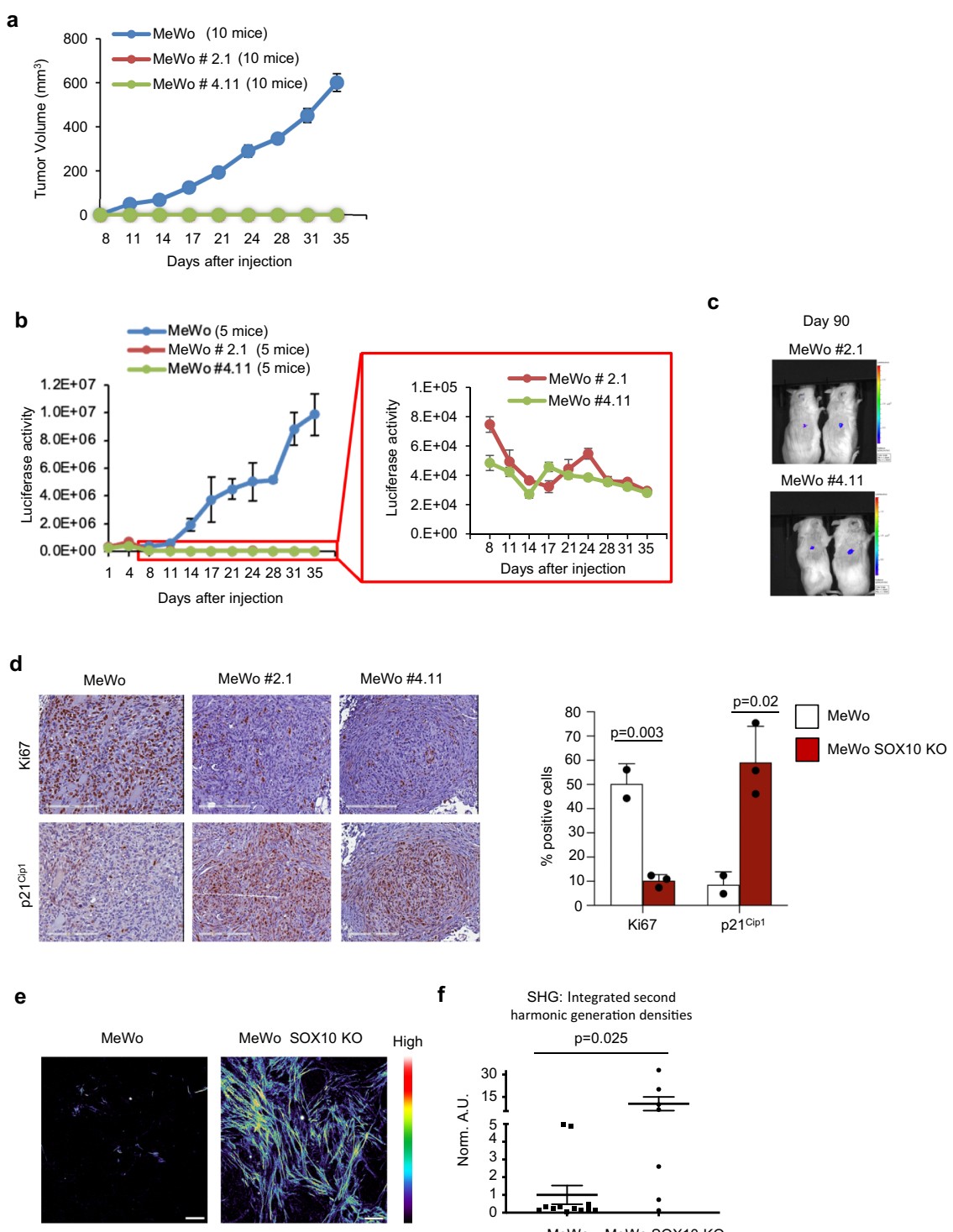

cells with the intent of targeting drug-tolerant persister cells. To this end, we screened a total of 1820 compounds using the anti-cancer library compound panel in parental and SOX10 knockout MeWo cells. Notably, all five cIAP1/2-XIAP inhibitors included in the screen (LCL161, birinapant, GDC0152, AZD5582, and BV6) effectively induced cell death in SOX10 knockout cells with little-to-no effect on parental cells (Fig. 6a). Embelin, a XIAP selective inhibitor, did not induce death in any of the cell lines analyzed (Fig. 6a). Thus, cIAP1 and/or cIAP2 may be relevant targets for inducing cell death in SOX10 knockout cells.

To validate our results, we focused on the cIAP1/2 inhibitor, birinapant, which is currently in clinical trials for head and neck squamous cell carcinoma (NCT03803774). We confirmed reduced expression of cIAP1/2 and XIAP following birinapant treatment by western blot analysis to show on-target effects (Supplementary Fig. 5A). In IncuCyte growth assays, birinapant reduced cell growth in SOX10 knockout cells compared to parental cells in a statistically significant manner, either at all doses analyzed for MeWo #2.1 or at higher doses (50 and 100 nM) for MeWo #4.11 (Fig. 6b and Supplementary Fig. 5B).

**Fig. 4 Loss of SOX10 induces a dormant/quiescent phenotype in vivo. a** Average tumor volume ±SEM for MeWo parental and SOX10 knockout clones #2.1 and #4.11. Number of mice per cohort as indicated in the figure. **b** Average Luciferase signal (Avg Radiance [p/s/cm²/sr]) ±SEM for MeWo parental and SOX10 knockout clones #2.1 and #4.11. Number of mice per cohort as indicated in the figure. **c** Luciferase signal in MeWo #2.1 (2 mice) and #4.11 (2 mice) 90 days after injection. Mice injected with parental MeWo cells did not survive 90 days; hence, their exclusion from the 90 days imaging. **d** IHC comparing p21$^{Cip1}$ and Ki67 expression in MeWo parental versus SOX10 knockout (clones #2.1 and #4.11) tumors collected at the end of the experiment (day 35). Shown is the mean ± SD from three independent tumors generated either from parental or SOX10 knockout MeWo cells. Scale bar, 200 μm. *p*-values were calculate using two-sided *t*-test. **e** Representative images showing second-harmonic generation signatures from three independent tumors from MeWo parental or SOX10 knockout xenografts. Images shown correspond to reconstituted monochromatic images pseudocolored according to "intensity heat-maps" of total second-harmonic generation signal. Warmer tones indicate higher second-harmonic generation signals (color tone bar is provided). Scale bars, 50 μm. **f** Quantitative analysis of polymerized collagen signatures from images in **e**. Three independent tumors generated either from parental or SOX10 knockout MeWo cells were analyzed. In total, 12 images from parental and 7 images from SOX10 knockout cells tumors were assessed. Values presented are normalized to MeWo parental xenograft. Data are expressed as mean ± SD. *p* = 0.025. *p*-values were calculated using the Wilcoxon two-sample test.

Birinapant elicited a similar, albeit less strong, effect in SOX10 knockout A375 cells (Supplementary Fig. 5C). A comparable trend was observed in parental and SOX10 knockout MeWo cells treated with GDC0152 and LCL161 (Supplementary Fig. 5D, E).

To identify the mechanisms involved in the differential response of SOX10-deficient versus -proficient cells to cIAP inhibitors, we analyzed the expression of IAP proteins. Strong upregulation of cIAP2 expression was detected in SOX10 knockout cells, while cIAP1 and XIAP levels were relatively unchanged (Fig. 6c). High cIAP2 levels were also observed in SOX10-deficient *NRAS* mutant melanoma cells, WM1361 and WM1366 (Supplementary Fig. 5F), which are known to be less sensitive to MEKi-induced cell death compared to other SOX10-proficient *NRAS* mutant cell lines[31]. To evaluate the clinical relevance of a SOX10-cIAP2 negative correlation, we analyzed a single-cell RNA-seq publicly available dataset obtained from patient-derived melanoma cell lines[11]. SOX10 expression was significantly associated with decreased BIRC3/cIAP2 RNA-seq counts (Fig. 6d and Supplementary Fig. 5G). Furthermore, a significant negative correlation was confirmed by analyzing the cutaneous melanoma mRNA expression TCGA dataset (Supplementary Fig. 5H).

Previous studies have suggested that low SOX10 expression characterizes the "undifferentiated" cell state[10], which has been associated with resistance to targeted therapy[2] and ICi[32]. Thus, we analyzed the expression of BIRC3 across different cell states expressing various levels of SOX10. RNA-seq data derived from human[10] and mouse[32] cell lines confirmed that BIRC3 mRNA levels are upregulated in the "undifferentiated", SOX10 low/negative state (Supplementary Fig. 5I, J). However, a similar upregulation was observed in the Neural Crest Stem Cell (NCSC)-like state, which expresses SOX10 (Supplementary Fig. 5I, J). Since the NCSC-like state retains a low level of MITF, this transcription factor may play a role, but further studies are necessary. Importantly, both "undifferentiated" and "NCSC-like" states have been associated with resistance to ICi and MAPKi, mirroring the cross-resistance observed in patients treated with immune or targeted therapy.

Increased expression of cIAP2, and to a lesser extent cIAP1, was observed in SOX10-deficient A375-derived CRT lines compared to parental A375 cells (Fig. 6e). Similarly, higher levels of cIAP2 were observed in 1205Lu-derived SOX10-deficient PBRT16 cells compared to parental 1205Lu cells (Supplementary Fig. 5K). SOX10-deficient CRT35 and CRT34 cells showed selective sensitivity to birinapant, starting at 0.5 and 0.1 μM, respectively, compared to parental A375 cells (Fig. 6f and Supplementary Fig. 5L, M). Interestingly, BRAFi + MEKi elicited a degree of protection to CRT cells in vitro against the effects of birinapant (Fig. 6f and Supplementary Fig. 5N, quantified in Fig. 5O).

Given the ability of SOX10-deficient cells to increase fibrous ECM protein deposition/organization (Figs. 3b, 4e, and 5g), we next queried whether birinapant treatment inhibits these alterations in ECM production and remodeling. A reduction of ECM protein deposition was observed in MeWo SOX10 knockout cells and SOX10-deficient A375 CRT35 cells treated with birinapant; however, it will be important to separate these effects from the ability of birinapant to induce cell death in SOX10-deficient cells (Supplementary Fig. 5P, Q).

Since birinapant has been utilized clinically, we tested its effects in combination with BRAFi+MEKi in A375 xenograft models. Co-treatment with birinapant significantly delayed the onset of BRAFi+MEKi resistance and improved mice survival (Fig. 6g,h and Supplementary Fig. 5R, S). The triple combination (BRAFi + MEKi+birinapant) versus BRAFi+MEKi caused more complete and durable responses and cleared tumors did not reoccur following drug removal. IHC analysis of SOX10 expression showed that BRAFi/MEKi/birinapant-treated tumors expressed a significantly higher level of SOX10 compared to the control tumors. (Supplementary Fig. 5T). These data support the notion that targeting the SOX10-negative subpopulation improves the durable efficacy of BRAFi+MEKi.

## Discussion

Given intratumor heterogeneity in cutaneous melanoma, an important issue is to characterize the states associated with phenotypic switching. Understanding the mechanistic switches to drug-tolerant subpopulations and selectively targeting them is likely to improve treatment for Stage III and IV melanoma. We focused on SOX10, a transcription factor that is readily expressed in melanoma lesions[33,34] and often used as a diagnostic marker[35]. Our findings demonstrate that SOX10 is heterogeneously expressed in treatment-naïve melanoma samples. Additionally, we found that SOX10 loss characterizes an invasive but dormant/quiescent-like phenotype that represents a phenotypic switch from the proliferative state of SOX10-expressing cells. The quiescent-like state is associated with tolerance to MAPK targeting agents but renders cells sensitive to cIAP1/2 inhibitors (Fig. 7). The cIAP1/2 inhibitor, birinapant, improves the durable efficacy of BRAFi+MEKi. Our studies underscore the importance of targeting SOX10-deficient subpopulations to prevent invasion and drug tolerance.

We show that the expression of SOX10 can direct melanoma cells towards a proliferative state and that its loss leads to invasive characteristics. Conversely, a previous study suggested that a reduction in SOX10 expression down-regulates invasiveness[15]. It is known that depletion of SOX10 reduces cell proliferation and induces senescence[36,37], and that Sox10 over-expression in zebrafish promotes melanoma formation[14]. Our study shows that SOX10-deficient melanoma cells are slow-cycling, which is an

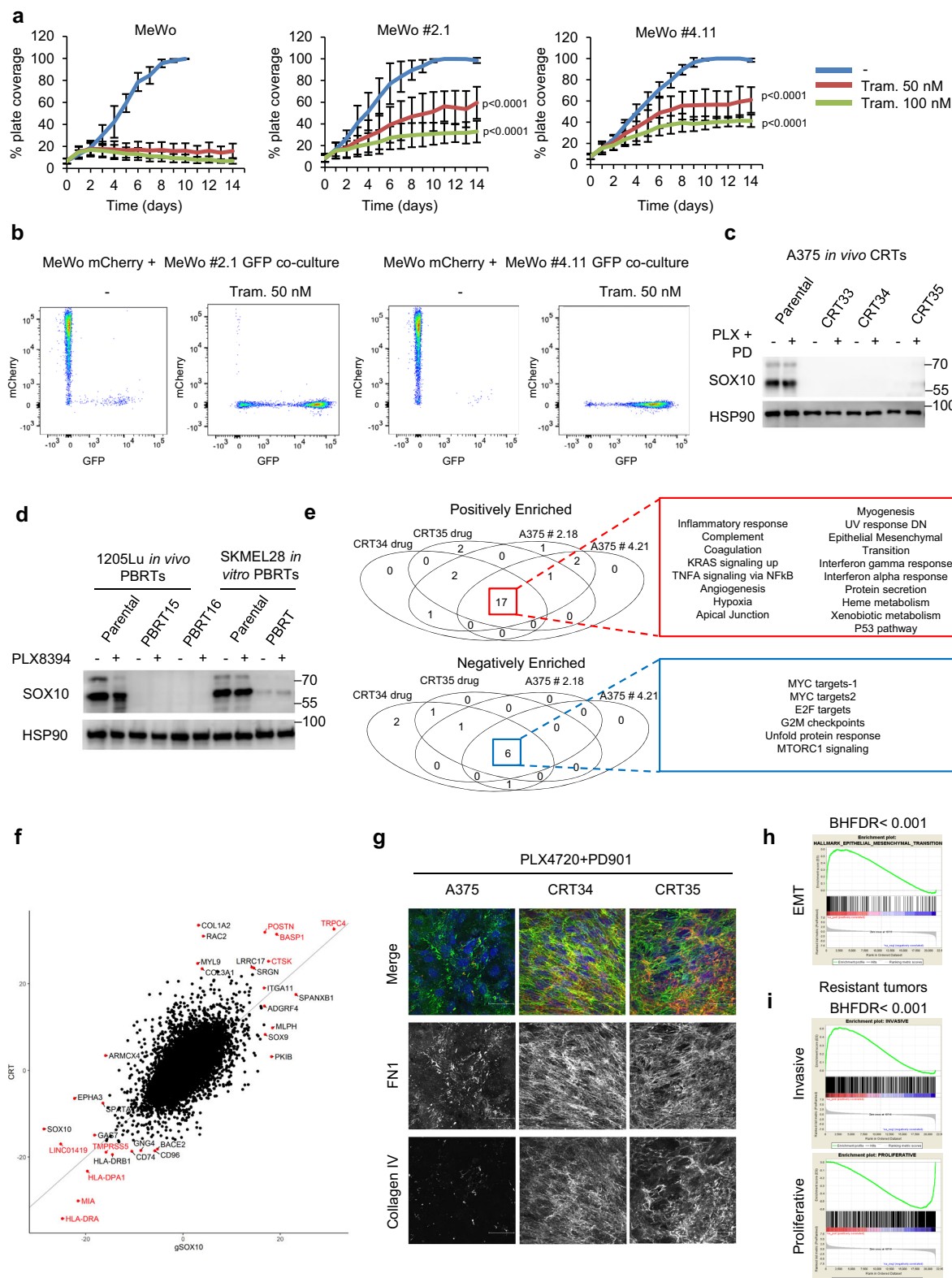

effect associated with reduced expression of MITF and cyclin D3 as well as enhanced p21[Cip1]. Other studies suggest that SOX10 may mediate effects on melanoma growth and survival via regulation of the long non-coding RNA, SAMMSON, and mitochondrial homeostasis[38], via control of SOX9[39] and/or through regulation of the small GTPase RAB7[40]. How these mechanisms, in combination with MITF loss, contribute to the slow-cycling phenotype awaits further examination. Furthermore, the invasive

phenotype of SOX10-deficient cells was associated with upregulated ZEB1, FN1, and FAK phosphorylation. ZEB1 is an EMT transcription factor known to induce ECM remodeling[41]. FN1 is often secreted by the mesenchymal compartment whereas collagen IV is commonly produced by tumor cells. FAK signals downstream of integrins which are activated by FN1, collagen IV, and other ECM proteins, all of which have been previously associated with resistance to the BRAFi, vemurafenib[42]. FAK inhibitors are being tested

**Fig. 5 SOX10 loss induces tolerance to MAPK targeting agents. a** Cells were treated with 50 or 100 nM trametinib. Treatment was renewed three times per week. Shown is the mean ± SD from three independent experiments. *p*-values were calculated using two-sided model-based *t*-test tests and adjusted for multiple testing using the Hochberg method and represent statistical analysis comparing growth inhibition of MeWo parental vs MeWo #2.1 and MeWo #4.11 from day 0 to day 10. *p*-values are shown. **b** mCherry-MeWo and GFP-MeWo SOX10 knockout #2.1 or #4.11 cells were co-mixed at the ratio of 2:1. Cells were co-cultured for 90 days in the presence or absence of 50 nM trametinib. Co-cultures were collected and analyzed by FACS for mCherry and GFP positivity. **c** Cells were treated with 1 μM PLX4720 plus 35 nM PD325901 for 24 h. Cells were lysed and western blotted as indicated. The experiment was repeated independently three times with similar results. **d** Cells were treated with BRAFi (500 nM PLX8394). Cells were lysed and western blotted as indicated. The experiment was repeated independently three times with similar results. **e** Venn diagrams showing the commonality of enriched gene sets (BHFDR < 0.05) from the Hallmark gene set collection (*n* = 50) for comparisons between A375 gSOX10 or CRTs in the presence of BRAFi+MEKi and parental control groups. **f** A scatter plot showing differential gene expression for gSOX10 and CRT groups over parental A375. Wald's test statistic was averaged across the gSOX10 #2 and gSOX10 #4, and the CRT34 and CRT35, conditions. Red dots and gene names indicate the top 10 most significantly up and downregulated genes for the CRT and gSOX10 groups, and genes that are common between CRT and gSOX10 are labeled in red. **g** Cells were treated with 1 μM PLX4720 + 35 nM PD0325901. Treatment was renewed every 48 h. Cells were stained for FN1 and collagen IV. The experiment was performed independently twice, and representative images are shown. Scale bars, 50 μm. **h** Enrichment plot of EMT (BHFDR < 0.001) comparing patient samples before and after MAPK targeting treatment from the Sun. et al. dataset. **i** Proliferative/invasive signature (Verfaille, et al.) for patients post vs pretreatment, controlled for patient (Sun, et al.).

clinically and may be useful for blocking the spread/growth of slow-cycling invasive cells. Changes in ECM fibrillogenesis, first noted by the Keely group[43], have been reported in a plethora of solid tumors[44,45] and were shown to be clinically relevant in melanoma[46,47].

Slow-cycling melanoma cells have been linked to drug tolerance[48]. We show that SOX10 deficiency is sufficient to cause MAPK targeted therapy tolerance and that BRAFi and/or MEKi treatment preferentially selects for SOX10-deficient cells. SOX10 loss is associated with melanomas that have acquired resistance to BRAFi[16]. Furthermore, SOX10-negative clones have been identified in the minimal residual disease of *BRAF*-mutant PDX models following BRAFi and MEKi and in 'on-treatment' patient samples[2]. The selective outgrowth of SOX10-deficient cells on MEKi suggests that acquired resistance to MAPK targeted therapy may arise from pre-existing SOX10-deficient cells. We show that SOX10-deficient/low cells are present to varying levels in treatment-naïve tumors. Whether the loss of SOX10 is driven by genomic alterations or reversible epigenetic events, as previously suggested[6,49], needs further elucidation. While our study focused on signaling pathway-targeted inhibitors, the role of SOX10 in response to immune-based therapies also needs evaluation. We note that both of the SOX10-negative melanomas from the single-cell sequencing dataset were ICi-resistant[20], and preclinical studies suggest that low levels of SOX10 are associated with a lack of response to ICi[32]. Additionally, analysis of publicly available datasets[10,32] indicates that "NCSC-like" and "undifferentiated" states, which have both been associated with resistance to targeted and immune therapies, express high levels of BIRC3 mRNA. A future direction will be to investigate the role of BIRC3 in tolerance/resistance to targeted therapies and immunotherapy, and whether cIAP2i can prevent or delay the onset of resistance for both therapies. Data from the literature suggest that cIAP2 can be regulated post-translationally[50,51], thus, it will be important to assess cIAP2 protein levels across cell states moving forward.

The properties of SOX10-deficient cells led us to identify novel therapeutic strategies to target this subpopulation with the goal of enhancing targeted therapy regimens. Broad genomic and pharmaceutical screens have identified synthetic lethal drug combinations and drug-gene connections that lead to the death of distinct cellular populations. A pharmacological inhibitor screen identified a class of cIAP1/2 inhibitors as selectively targeting SOX10-deficient cells, an effect likely due to a dramatic up-regulation of cIAP2 proteins. The cIAPs, a family of structurally related proteins, including XIAP (X-linked IAP), cIAP1, cIAP2, and survivin, are known to inhibit apoptosis by blocking the activation of effector caspases. IAPs inhibitors (also known as

SMAC mimetics) cause the degradation of cIAP1 and 2[52]. Previous studies have shown that birinapant inhibits the growth of many melanoma cell lines if combined with TNF-α treatment[21]. In preclinical models, birinapant synergizes with TRAF2 inactivation to induce tumor volume reduction and extend survival in mice, likely impacting T cell functionality[53]. Addition of anti-PD1 to this combination induced superior tumor control and improved overall survival in mouse models.

Multiple monovalent, as well as bivalent, cIAP1/2 inhibitors have entered clinical cancer trials. In general, these compounds are well tolerated and have a reasonable safety profile; however, they have shown limited clinical activity as single agents[54]. Previous studies showed synergistic effects of birinapant in combination with BRAFi in BRAF V600E colorectal cancer cells[55]. Consistently, our data suggest that adding cIAP1/2 inhibitor to the regimen of BRAFi and MEKi will delay the onset of acquired resistance.

An interesting aspect of this strategy is that, at least in vitro, BRAFi+MEKi elicited a protective effect on SOX10-deficient cells treated with birinapant. Previous studies have suggested that BRAFi-resistant cells develop drug addiction rendering them sensitive to drug withdrawal[29]; however, results from clinical trial data with BRAFi+MEKi indicate that continuous treatment regimens are more effective[56]. Further studies are needed to identify optimal scheduling of birinapant with BRAFi+MEKi combinations to maximize tumor regression durability and minimize potential toxicity. In addition, investigating cIAP1/2 inhibitor effects on the tumor immune microenvironment may provide insights into how these agents might potentially interact with ICi. Together these findings raise several potential strategies for the incorporation cIAP1/2 inhibitors into treatment regimens for cutaneous melanoma.

## Methods

**Cell culture.** MeWo cells (kindly donated by Dr. Barbara Bedogni, when at Case Western Reserve, Cleveland, OH in 2014), A375 parental cells (purchased from ATCC in 2005), and vehicle-treated xenograft derived A375 cells (A375 CTL) were cultured in DMEM with 10% FBS. A375 BRAFi+MEKi combination tolerant/resistant cells (CRT33, CRT34, and CRT35, previously labeled as CRT13, CRT14, and CRT15[1]) were cultured in the presence of PLX4720 (1 μM) and PD325901 (35 nM). 1205Lu and 1205LuTR GAL4-ELK1 reporter cells were modified from the parental line (generated by Dr. Meenhard Herlyn, The Wistar Institute, Philadelphia, PA in 2005). 1205Lu, paradox breaker resistant tumor (PBRT) #15 and #16 cells[25], and SKMEL28 cells (purchased from ATCC in 2002) were cultured in MCDB153 (Sigma) with 2% FBS, 20% Leibowitz L-15 medium, and 5 μg/ml insulin. 1205Lu-PBRT #15 and #16 and SKMEL28-PBRT cells were cultured in 0.5 μmol/L PLX8394. *BRAF* and *NRAS* mutation status in cell lines was validated by Sanger sequencing. Cells were assayed for mycoplasma contamination every two months with MycoScope Kit (Genlantis). Short-tandem repeat analysis was completed for parental and CRISPR SOX10 knockout MeWo cells and the respective

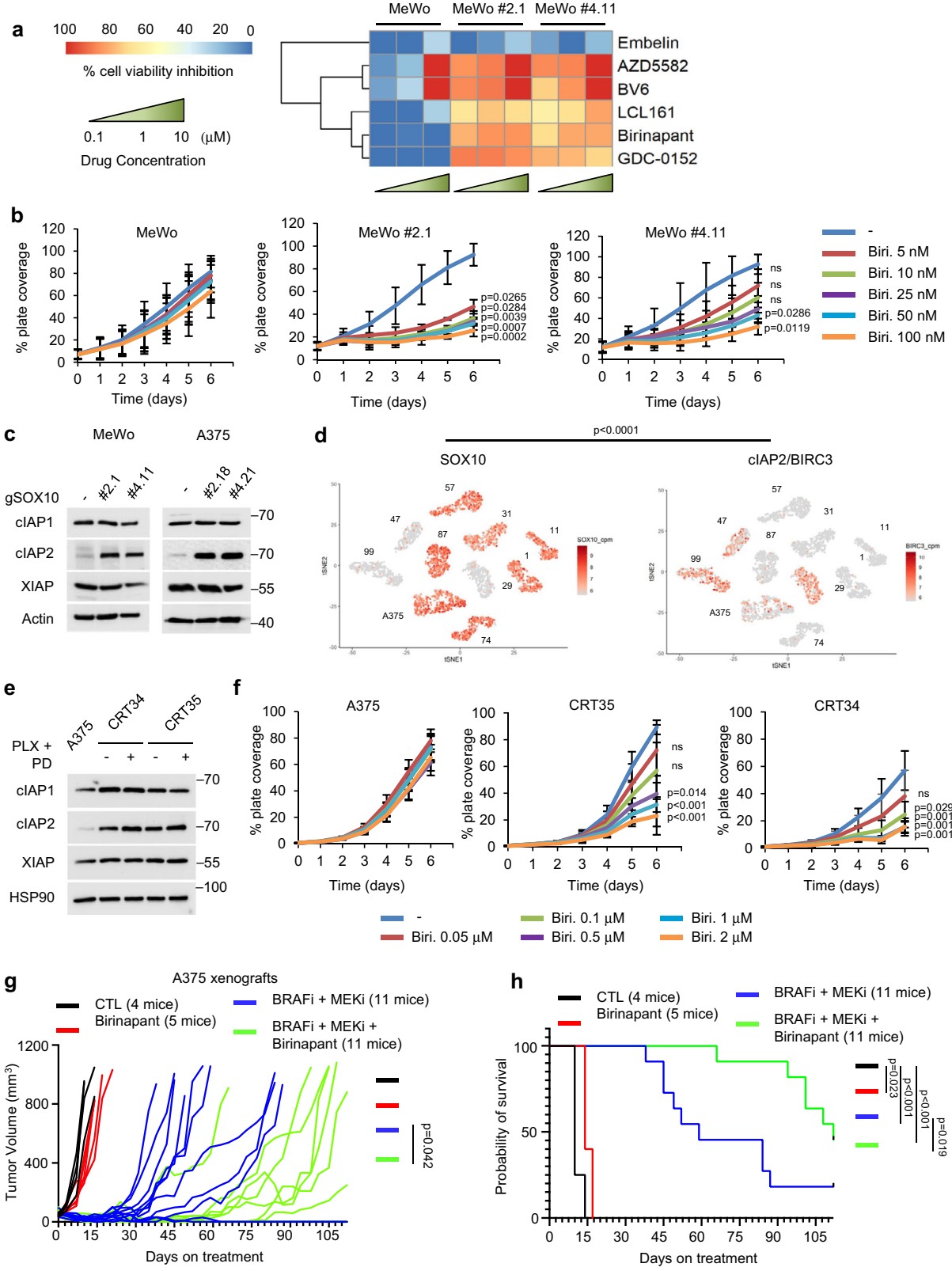

luciferase, mCherry and GFP labeled versions (April 2019), for parental, CRISPR SOX10 knockout, and tolerant/resistant A375 cells (October 2020), for 1205Lu (January 2021) and for 1205LuTR GAL4-ELK1 reporter and PBRT cells (February 2017). All cell lines matched known profiles. Cells were cultured at 37 °C with 5% $CO_2$ in a humidified chamber.

**CRISPR**. To generate CRISPR SOX10 knockout cells, MeWo and A375 cells were co-transfected with Edit-R Cas9 plasmid harboring puromycin-resistance marker

(Horizon Discovery, U-005100-120), two different crRNAs for SOX10 (guide#02, Horizon Discovery CM-017192-02-0002, target sequence GATGGT-CAGAGTAGTCAAAC, and guide#04, Horizon Discovery CM-017192-04-0002, target sequence GTCCAACTCAGCCACATCAA), and tracrRNA using Dharma-FECT Duo (Horizon Discovery). After 48 h, puromycin was added to the medium, and cells were selected for growth. Individual clones were expanded and tested for SOX10 knockout by western blotting. To generate the MeWo SOX10 knockout pool, clones #2.1, # 2.2, # 2.8, and #4.11 were co-mixed at equal ratios.

**Fig. 6 Synthetic lethality of IAP inhibitors towards SOX10-deficient population. a** Synthetic lethality of IAP1/2-XIAP inhibitors toward the SOX10-deficient population expressed as inhibition of cell viability. **b** The same number of cells were seeded for each cell line. Cells were treated with increasing concentrations of birinapant. Treatment was renewed every 48 h. Shown is the mean ± SD from three independent experiments. *p*-values were calculated using two-sided model-based *t*-test tests and adjusted for multiple testing using the Hochberg method and represent statistical analysis comparing birinapant-induced growth inhibition in MeWo parental vs MeWo #2.1 and MeWo #4.11. *p*-values are shown. **c** Cell lysates were western blotted, as indicated. The experiment was repeated independently three times with similar results. **d** scRNA-seq cell line data for SOX10 and cIAP2/BIRC3. *p* < 0.001 for zero-inflation model and *p* = 0.002 for mean model. **e** Cells were treated with PLX4720 (1 μM) + PD0325901 (35 nM) for 24 h, lysed and western blotted. The experiment was repeated independently three times with similar results. **f** The same number of cells were seeded for each cell line. Cells were treated with increasing concentrations of birinapant. Treatment was renewed every 48 h. Shown is the mean ± SD from three independent experiments. *p*-values were calculated and adjusted as in **b**. and represent statistical analysis comparing birinapant-induced growth inhibition in A375 parental vs CRT35 or CRT34 cells. *p*-values are shown. **g** A375 xenograft tumor growth, day 0 corresponding to the first day of treatment. Mice were treated 200 PPM PLX4720 plus 7 PPM PLX2695 and/or injected with 100 μL (for female mice) or 150 μL (for male mice) birinapant solution (3 mg/ml). AIN-76A diet was used as vehicle. Number of mice per cohort as indicated. Statistical significance was calculated as the time to tumor regrowth (tumor volume >100 mm$^3$) and corresponding median survival times were estimated using the Kaplan–Meier method. The two-sided log-rank test was used to compare the time to regrowth between treatment groups. *p*-value is shown. **h** Mouse survival curve for the in vivo experiment shown in **g**. *p*-values were calculated using the two-sided log-rank tests and were adjusted for multiple testing to control for the False Discovery Rate (FDR) using the method of Benjamini and Hochberg. *p*-values are shown.

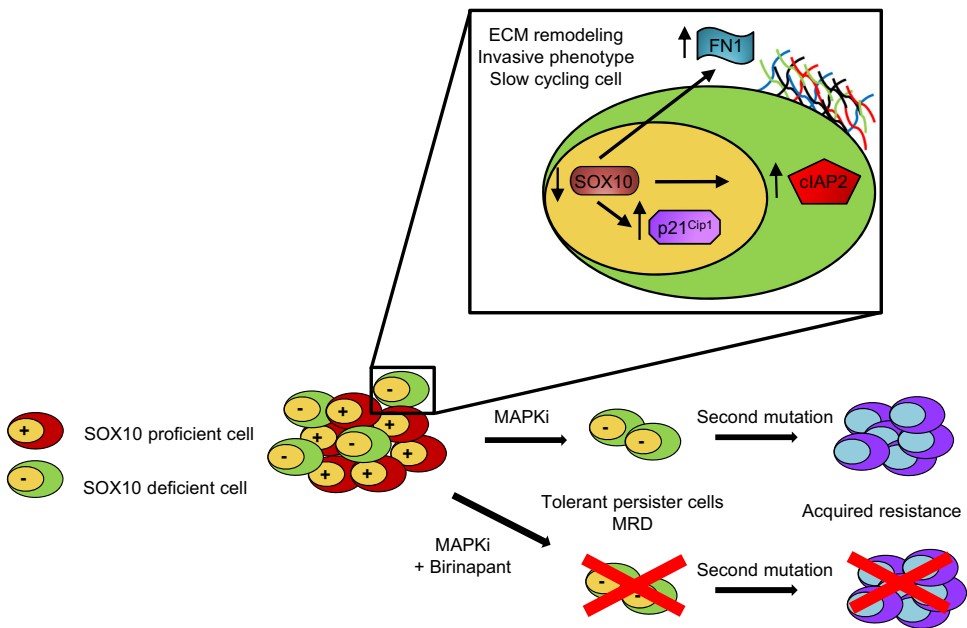

**Fig. 7 Schematic showing the effects of SOX10 loss in melanoma.** SOX10 loss leads to an invasive but dormant-like/quiescent phenotype associated with ECM remodeling and tolerance to MAPK targeting agents. SOX10-deficient cells show upregulation of cIAP2 that is associated with sensitivity to cIAP1/2 inhibitors.

**Generation of luciferase and fluorescent-labeled cells**. For luciferase labeling, MeWo parental and CRISPR SOX10 knockout cells were transduced with pLenti-puro TO/V5-GW firefly luciferase plasmid and selected for using puromycin. For mCherry labeling, MeWo cells were transduced with pLV-mCherry (Addgene #36084), and highly expressing cells were sorted. A similar approach was followed for GFP labeling of MeWo CRISPR knockout SOX10 #2.1 and #4.11 cells, except that the pLV-eGFP (Addgene #36083) plasmid was used.

**Inhibitors**. Dabrafenib (BRAF inhibitor), PD0325901 (MEK inhibitor), trametinib (MEK inhibitor), and birinapant were purchased from Selleck Chemicals. PLX4720 (BRAF inhibitor), PLX2695 (MEK inhibitor), and PLX8394 (BRAF inhibitor) were provided by Dr. Gideon Bollag (Plexxikon Inc., Berkeley CA).

**Single-cell sequencing analysis**. Single-cell RNA-Seq data for seven treatment-naïve and seven immune checkpoint inhibitor-resistant melanoma patient tumors originated from Jerby-Arnon, et al.[20] These included tumors with at least 50 malignant cells. The Single Cell Portal (https://portals.broadinstitute.org/single_cell/study/melanoma-immunotherapy-resistance) was used to generate tSNE plots of malignant cells. A loom file containing scRNA-seq data of 10 cutaneous melanoma cell lines from Wouters et al.[11] was obtained from http://scope.aertslab.org/#/Wouters_Human_Melanoma. SCopeLoomR (v0.10.2 https://github.com/aertslab/SCopeLoomR) was used to load preprocessed cell line data

into R (v4.0.2 https://www.R-project.org/). Pseudo-counts per million data were calculated from raw counts using the edgeR package (v 3.30.3)[57]. Normalized counts and tSNE coordinates were used to plot gene expression levels of each cell. Scatter plots were generated using the ggplot2 package (v 3.3.2 https://ggplot2.tidyverse.org).

**Bulk RNA-seq sample prep, data acquisition, and analysis**. For MeWo CRISPR SOX10 and parental cell line samples, total RNA (100 ng) was used to prepare libraries using TruSeq Stranded Total RNA kit (Illumina, CA) by following the manufacturer's protocol. The final libraries (at the concentration of 4 nM) were sequenced on NextSeq 500 using 75 bp paired-end chemistry. For A375 CRISPR SOX10, parental, and CRT samples, 200 ng aliquot of each sample was transferred into library preparation, which uses an automated variant of the Illumina TruSeq™ Stranded mRNA Sample Preparation Kit. The final libraries were sequenced on Illumina NovaSeq 6000 using 101 bp paired-end with an eight-base index barcode read. Raw FASTQ sequencing reads for three patients with pretreatment and post MAPK pathway inhibitor resistance tumor samples were obtained from the Sequence Read Archive under the accession number SRP029434 using the SRA toolkit (v 2.10.4)[58].

For each dataset above, raw FASTQ sequencing reads were mapped against the reference genome of Homo sapiens (Ensembl Version GRCh38.p12). Further information was utilized from the gene transfer format annotation by GENCODE (v28 or v30) using RSEM (v1.2.28)[59]. Total read counts and normalized

Transcripts Per Million (TPM) were obtained using RSEM's calculate-expression function. Batch effects and sample heterogeneity were tested for using iSeqQC (v1.0 https://github.com/gkumar09/iSeqQC)[60]. Differential gene expression between paired-sample patient tumors or guide RNA and control samples were tested for using the DESeq2 package (v1.28.1)[61]. Genes were considered differentially expressed (DE) if they had an adjusted p ≤ 0.05 and an absolute fold change ≥2. Gene Set Enrichment Analysis (GSEA v3.0 or v4.0.1)[62], was performed to identify significantly altered pathways. The MSigDB Hallmark gene set collection (v6.2 or v7.0)[63], GO Biological Process gene sets[64], and invasive and proliferative signatures from Verfaillie et al.[6] were used for analyses. The DESeq2 test statistic was used as a ranking metric to perform GSEA in pre-ranked mode, with genes having zero base mean or "NA" test statistic values filtered out to avoid providing numerous duplicate values to GSEA. GSEA pre-ranked analysis was performed using the "weighted" enrichment statistic. The number of permutations was set to 1,000, and FDR q-values equaling zero are reported as less than 0.001. Venn diagrams, heatmaps and scatter plots were generated using the VennDiagram (v1.6.20 https://CRAN.R-project.org/package=VennDiagram), pheatmap (v1.0.12 https://CRAN.R-project.org/package=pheatmap) and ggplot2 (v3.3.2 https://ggplot2.tidyverse.org/) packages in R (v3.5.1 or v4.0.2 https://www.R-project.org/). The RNA-seq data generated for this publication can be found under NCBI BioProject numbers PRJNA701949, PRJNA748713, and PRJNA748714.

**Immunofluorescence**. A375, MeWo, and 1205Lu cells were grown on coverslips overnight, washed with PBS, and fixed with 4% formaldehyde for 10 min. Fixed coverslips were washed three times with PBS and incubated with 2% BSA/0.2% Triton X-100 in PBS for 30 min at room temperature to permeabilize cells and block nonspecific staining. Coverslips were then incubated with primary SOX10 antibody (Abcam [SP267], ab227680, 1:200) diluted in 2% BSA/0.2% Triton X-100 in PBS overnight at 4 °C. After washing three times in PBS, coverslips were incubated for 1 h at room temperature with Alexa Fluor 488 (Invitrogen, A-11034, 1:1000) conjugated secondary antibody, Phalloidin-TRITC (Sigma, P1951, 1:500) to visualize the cytoplasm, and DAPI (Abcam, ab228549, 1:1000) to visualize nuclei. Coverslips were mounted and imaged using a 40X objective lens on a Nikon A1R Microscope with NIS-Elements AR software.

**Immunohistochemistry**. Deidentified patient samples were collected from Thomas Jefferson Hospital under an IRB-approved protocol (#10D.341) which includes written informed consent and was in accordance with recognized ethical guidelines. Sections from formalin-fixed and paraffin-embedded (FFPE) tumors were stained with SOX10 antibody (Abcam [SP267], ab227680, 1:200). Selected areas of SOX10-deficient cells were evaluated by two pathologists: Dr. Hookim (Thomas Jefferson University) and Dr. Xu (University of Pennsylvania) and identified as cancer cells. Tissue microarrays (TMA) were generated at the University of Pennsylvania and information about the tumor samples was obtained from Krepler, et al.[21]. The TMA contained 13 Stage III melanoma samples, 1 Stage III/IV melanoma sample, and 38 Stage IV melanoma samples. Stage information was not available for 6 of the samples. Tumors were stained with SOX10 antibody (Abcam [SP267], ab227680, 1:200). Samples were derived from 64 melanoma patients and each patient sample was present on the TMA in duplicate. However, two patient samples (WM3901 and WM4115) were excluded from the analysis because the tissue was no longer suitable for IHC. In each sample, the percentage of cancer cells was classified, based on SOX10 expression, as negative (intensity = 0), low (intensity = 1), medium (intensity = 2) and high (intensity = 3) by Dr. Xu. The H score for each tumor was calculated according to the formula: H-score = intensity 1 x area + intensity 2 x area + intensity 3 x area. For each patient the H-score was calculated as the average of the sample duplicates. For nine of the tumors analyzed, only one replicate was deemed suitable for analysis.

For xenograft staining, MeWo, MeWo CRISPR SOX10 #2.1, and MeWo CRISPR SOX10 #4.11 tumors were harvested from mice at 35 days post-injection. Tissues were processed for FFPE and stained with anti-Ki67 (Abcam, ab16667, 1:200), anti-p21[Cip1] (Cell Signaling, #2947, 1:200), and anti-SOX10 (Abcam [SP267], ab227680, 1:200) antibodies. The percentages of Ki67 and p21[Cip1] positive cells were evaluated by a pathologist (Dr. Xu). Additional information regarding the IHC protocol is included in Supplementary Methods.

**Anti-cancer library compound panel drug screen**. Cell viability screening against the MCE anti-cancer library (1820 compounds) was performed using CellTiterGlo® (Promega). Both control and SOX10 knockout cells were maintained in complete media, trypsinized, and plated at 500 cells/well in 40 µL of complete media the day before the experiment in white, clear bottom 384-well plates. 50 nL of test compound was added to each well using the Janus MDT Nanohead (Perkin Elmer). Each compound was screened at a final concentration of either 10, 1, and 0.1 µM. After a 72 h incubation at 37 °C + 5% CO₂, 20 µL of CellTiterGlo reagent was added to each well. After 15 minutes, luminescence was measured using the Envision Multi-mode plate reader (PerkinElmer). The raw data were normalized to % inhibition, where 0% is the RLU in the presence of DMSO only, and 100% is the RLU in the presence of 1 µM bortezomib. Estimated IC50 values for each compound were determined using nonlinear regression fits on the data to a one-site binding model in XlFit (IDBS). Because only 3 data points were used in

this calculation, the top and bottom of the curve were fixed to 100, and 0%, respectively, with a constant slope value of 1.

**ECM fibers orientation**. Tri-dimensional monochromatic z-plane stacked images were two-dimensionally reconstituted to maximal projection images of FN1 and collagen IV using FIJI software[65] (ImageJ 1.53f51, https://imagej.net/software/fiji/). Aided by the same software, an initial screening evaluated the percentage of fibers area coverage (% FAC) per image. For this, a fixed threshold based on grayscale pixel intensity was used to discriminate ECM fiber structures from disperse cytosolic signal within the cells. An arbitrary cutoff value of 20% FAC was set and only images above this value (see Supplementary Fig. 2C) were processed for fiber orientation analysis. Using FIJI's plugin OrientationJ (Biomedical Image Group, http://bigwww.epfl.ch/demo/orientation/), as described before[66,67], a comprehensive orientation analysis[68] was conducted using similar plugin settings for all the images evaluated. In brief, the mode angle calculated for each image was set to 0° and the rest of angle readouts normalized accordingly, followed by the adjustment of angle fluctuations between a range of −90° to 90°. The percentage of fibers oriented within −15° and 15° from the mode angle was determined for each normalized set, where greater percentages reflect higher levels of fibers organization.

**Microscopy and analysis of second-harmonic generation (SHG) of polarized light**. Tissue specimens processed for FFPE were sliced (~4 µm), mounted on microscopy slides, deparaffinized using xylene, rehydrated with progressive ethanol to water dilutions, rinsed in PBS, and redehydrated in progressive ethanol concentrations. Specimens were clarified in toluene before being mounted using Cytoseal-60 (Thermo Fisher Scientific, Waltham, MA), and then cured overnight in the dark prior to imaging. As described[69], SHG imaging was conducted in a Leica SP8 DIVE confocal/multiphoton microscope system (Leica Microsystems, Inc., Mannheim, Germany) equipped with a 25X HC FLUOTAR L 25x/0.95NA W VISIR water-immersion objective. Samples were illuminated at 850 nm with an IR laser Chameleon Vision II (Coherent Inc., Santa Clara, CA) and backward emission settings were used to collect second-harmonic generation signals, via a non-descanned detector that was configured to register between 410 and 440 nm wavelengths. Regions of interest were selected by a pathologist (Dr. Xu) who had knowledge of the sample's condition but was blinded to the expected result. Using the automated Leica Application Suite X 3.5.5 software, each of these areas was acquired by the collection of 2–4 regions, using identical settings and recorded as monochromatic 16-bit TIFF images, via stacks containing an average of 5.5 images (1 µm distance between each z-plane).

Image processing and digital analyses were conducted using FIJI software[65]. In brief, raw z-image stacks were reconstituted as two-dimensional maximal projections images. Signal to noise thresholds was set identical for all images and positive-signal areas were used to calculate integrated second-harmonic generation densities (e.g., SHG signal/SHG area). Data were normalized to the MeWo parental second-harmonic generation integrated density mean value. Results represent arbitrary units compared to control tissues.

**Western blot analysis**. Proteins were extracted with Laemmli sample buffer, resolved by SDS-PAGE, and transferred to PVDF membranes. Immunoreactivity was detected using HRP-conjugated secondary antibodies (CalBioTech, Spring Valley, CA) and chemiluminescence HRP-recognizing substrates (Thermo-Scientific, Waltham, MA) on a VersaDoc Multi-Imager. Primary and secondary antibodies used are listed in Supplementary Methods. Uncropped blots are supplied in the Source Data file.

**Cell growth assay (IncuCyte)**. Cells were plated in single wells of six-well plates. Cell growth was analyzed for percent plate coverage with IncuCyte Live Cell Analysis System. Forty-nine pictures per well were acquired every 24 h.

**Scratch-wound assay**. Cells were plated in single wells of IncuCyte ImageLock-96-well plates. The next day, cells were treated with mitomycin (1 µg/mL) for 2 h before the monolayer was scratched using the IncuCyte WoundMaker tool. Cells were washed with PBS and serum-free media (SFM) with 1% BSA was added to each well. Wound healing was analyzed by wound width (µm) using the IncuCyte Live Cell Analysis System. IncuCyte ScratchWound analysis software was used to take two pictures every 12 h per well. Significance was calculated by comparing CRISPR SOX10 knockout cells to parental cells at either 24, 48, or 72 h.

**ECM protein staining**. MeWo parental and MeWo SOX10 knockout cells (5 × 10⁵), CRT cells (3 × 10⁵), and A375 parental cells (1.25 × 10⁵) were plated on coverslips coated with 0.2% bovine gelatin and treated with ascorbic acid (50 µg/ml) every 48 h for 6 days. Cells were processed as previously described[68,70]. Briefly, samples were washed with DPBS + and fixed/permeabilized for 20 min using 4% paraformaldehyde with 5% sucrose and 0.1% Triton X-100 added. Samples were then washed twice with DPBS- with 0.05% Tween-20 and blocked for 1 h at room temperature in Odyssey Blocking Buffer containing 1% donkey serum. Samples were stained with FN1 (Sigma, #F3648, 1:200) and collagen IV

(Abcam, ab86042, 1:75) antibodies diluted in Odyssey Blocking Buffer for 1 h at room temperature. After three washes with DPBS- containing 0.05% Tween-20, samples were incubated with secondary antibodies (AlexaFluor-488 Invitrogen A-11034 1:1000 and AlexaFluor-594 Invitrogen A-11032 1:1000 or AlexaFluor-647 Invitrogen A-21236 1:1000) for 30 min at room temperature. Nuclei were stained with DAPI (Abcam, ab228692, 1:1000). After washing three times in DPBS- with 0.05% Tween-20, coverslips were mounted and pictures were taken with A1R Nikon confocal Ti-Eclipse inverted microscope (Nikon, Melville, NY) using NIS-Elements software.

**3D spheroid**. 5000 cells/well were plated in a 96-well plate coated with 1.5% agarose and were grown in suspension for five days to form spheroids. Spheroids were harvested and implanted into 3D collagen as previously described[31]. Briefly, spheroids were collected from 96-well plates using a transfer pipette and allowed to settle for 1 h. A 24-well plate was coated with a collagen mixture (rat tail Collagen Type I containing Reconstitution Buffer, Ham's F-12 Nutrient Mix, and 5% FBS) and incubated for 20 min at 37 °C. Spheroids were resuspended in a collagen mixture, plated on the coated 24-well plates, allowed to set for 20 min at 37 °C, and covered with a normal culture medium. After 4–7 days, spheroids were stained with Calcein AM (Invitrogen C1430) diluted in PBS for 1 h before bright-field images were taken with a C2 Nikon confocal Ti-Eclipse inverted microscope (Nikon) using NIS-Elements.

**Flow cytometry**. GFP or mCherry expressing MeWo cells ($3 \times 10^5$) were plated and left to incubate overnight. The next day, cells were trypsinized, spun down, washed twice with PBS, and resuspended in FACS Buffer (PBS with 1% FBS and 0.05% sodium azide). Cells were strained through a 70 μm cell strainer and transferred to flow tubes. Cells were run on a BD FACSCelesta (BD Biosciences; Franklin Lakes, NJ) and FACS data were analyzed using Flowjo software (v10.6.1). FSC-A/SSC-A plots were used to gate out cell debris. Then, the subsequent population was gated on FSC-A/FSC-H to remove doublets. The resulting population was assessed for GFP and mCherry expression.

**In vivo studies**. Animal experiments were performed at a Thomas Jefferson University facility that is accredited by the Association for the Assessment and Accreditation of Laboratory Animal Care. The Institutional Animal Care and Use Committee at Thomas Jefferson University approved these studies (Protocol #: 01052). All animals were provided with food and water ad libitum, and housed in cages (with a maximum of 5 mice/cage) in a temperature and humidity-controlled environment. Animals were maintained in housing conditions that allow for normal species behavior to minimize the development of abnormal behaviors and have access to humane and veterinary care. Three million MeWo parental or CRISPR SOX10 knockout cells (#2.1 and #4.11) were injected intradermally into the backs of NOD.Cg-Prkdc scid Il2rgtm1Wjl/SzJ (NSG) mice. Digital caliper measurements of the tumors were taken twice per week, and tumor volumes were calculated using the formula: volume = (length × width$^2$) × 0.52. In vivo bioluminescence detection was conducted using the Caliper IVIS Lumina-XR System (Caliper Life Sciences), and data acquisition was conducted using LivingImage Version 4.0 software. For firefly luciferase, mice were imaged after 10 min of intraperitoneal injection of d-luciferin (150 mg/kg).

For the A375 in vivo xenograft study, one million cells were injected intradermally into the backs of athymic mice (NU/J, homozygous, Jackson, 6–8 weeks, 20–25 g). When tumors were palpable, mice were randomly sorted into four cohorts. Mice were treated with BRAFi + MEKi (PLX4720 200 PPM + PLX2695 7 PPM) and/or injected intraperitoneally, twice per week, with 100 μL (for female mice) or 150 μL (for male mice) birinapant solution (formulated at 3 mg/ml in 5% DMSO and 15% captisol, CyDex Pharmaceuticals, aqueous solution). AIN-76A diet was used as vehicle. To prevent weight loss, after 14 days of treatment, the mouse diet was supplemented with ClearH$_2$O DietGel® Recovery and bacon softies (Bio-Serv) one day per week. At the end of the experiment the mice that did not present tumors ($n = 2$ for BRAFi+MEKi and $n = 4$ for BRAFi +MEKi+birinapant cohorts) were removed from treatment and observed for an additional 30 days. None of the mice developed tumors following treatment removal. Digital caliper measurements of the tumors were taken twice per week, and tumor volumes were calculated using the formula: volume = (length × width$^2$) × 0.52. Mouse weights were monitored once a week.

**Statistical analysis**. For scratch-wound assays, statistical analysis was performed using Student's two-sample t-tests, assuming unequal variance. For the 3D spheroid assay, the area of outgrowth and the core were compared between each MeWo SOX10 knockout and parental cells using the two-sided one-sample t-test of the null hypothesis. For analysis of second-harmonic generation (SHG) of polarized light (ECM fibers orientation), Wilcoxon two-sample test was used for two-group comparison. The Hodges-Lehmann estimate of location shift with the corresponding exact 95% confidence interval was computed to estimate the difference in integrated second-harmonic generation densities between MeWo and MeWo SOX10 knockout. For the IncuCyte experiments comparing parental and SOX10-deficient cells (CRISPR knockouts and CRTs), statistical analysis was performed as described: for each plate (replicate within treatment and dose) the log-transformed net confluency increase (lnNCI) at each day (D) (lnNCI(D)) from day 1 to the end

of the experiment was defined as the difference between log-transformed confluence at day D and log-transformed confluence at day 0. Log-transformed increases (for day ≥ 1) were analyzed in a longitudinal linear mixed-effects model with the fixed effects of genotype, treatment (if multiple drugs), dose, time, and random effect of the plate. Serial correlation between repeated over time measures in the same plate were modeled using autoregressive covariance structure. The dependence on time was modeled with a quadratic function. The fitted models were used to estimate the geometric mean proportion of growth inhibition (mPGI) for each treatment, non-zero treatment dose, and each genotype as the exponentiated mean difference between mean lnNCI (treatment, dose=D, geno) and meanNCI (treatment, dose=0, geno). The mPGI was compared between each clone and WT separately for each treatment and non-zero dose D by computing the PGI ratio (exponentiated mean difference in lnNCI). The p-values for the two-sided model-based t-test tests were adjusted for multiple testing using the Hochberg method.

Statistical analysis for single-cell RNA-seq data was performed using separate zero-inflated negative binomial (ZINB) regression models that model single-cell RNA-seq counts of BIRC3 as dependent on SOX10 counts and the total RNA-seq counts per cell as an exposure. The ZINB regression model includes a mean model for the negative binomial mean of BIRC3 count as dependent on SOX10 counts, and a zero-inflation probability model, which is essentially a logistic regression predicting the odds of zero counts as dependent on SOX10 counts.

For A375 in vivo studies, statistical analysis was performed to evaluate if the BRAFi+MEKi+birinapant triple combination significantly delayed the onset of resistance in comparison to BRAFi+MEKi combination. Statistical significance was calculated as the time to tumor regrow (tumor volume > 100) from day 0 and corresponding median survival times were estimated using the Kaplan–Meier method. The log-rank test was used to compare the time to regrow between treatment groups. For the mouse survival curve the time to sacrifice (in days) and corresponding median survival times were estimated using the Kaplan–Meier method. The log-rank test was used to compare the time to sacrifice between treatment groups (i.e., global comparison and comparison between control and each of 3 Tx group). The corresponding 95% CIs were also reported. In addition, six pairwise comparisons of the time to sacrifice were conducted with log-rank tests. p-values from these pairwise comparisons were adjusted for multiple testing to control the false discovery rate using the method of Benjamini and Hochberg. All the analyses were performed with SAS 9.4 (SAS Institute Inc., Cary, NC).

**Reporting summary**. Further information on research design is available in the Nature Research Reporting Summary linked to this article.

## Data availability

The RNA-seq data generated from the MeWo gSOX10, A375 gSOX10 and CRT, and 1205Lu TR PBRT samples are available under NCBI BioProject numbers PRJNA701949, PRJNA748713, and PRJNA748714, respectively. Human and mouse reference genomes (GRCh38.p12 and GRCm38.p6) and gene and transcript annotation data (v28, v30, and M25) were obtained from GENCODE (https://www.gencodegenes.org/). MSigDB Gene Set Collections (Hallmark and GO Biological Process, v6.2 and v7.0) were obtained from https://www.gsea-msigdb.org/. Invasive and proliferative signatures were obtained from https://static-content.springer.com/esm/art%3A10.1038%2Fncomms7683/MediaObjects/41467_2015_BFncomms7683_MOESM1477_ESM.xlsx. Single-cell RNA-Seq data for malignant cells from 7 treatment-naïve and 7 immune checkpoint inhibitor-resistant melanoma patient tumors were originated from Jerby-Arnon, et al. 2018[20]. Normalized expression data for a single gene are publicly available to view in tSNE space using the Single Cell Portal (https://portals.broadinstitute.org/single_cell/study/melanoma-immunotherapy-resistance). Cell annotation and normalized expression data are freely available for bulk download from the Single Cell Portal (https://singlecell.broadinstitute.org/single_cell/) after registering for an account using a Google-managed identity (https://singlecell.broadinstitute.org/single_cell/terms_of_service). A.loom file containing scRNA-Seq data for 10 melanoma cultures was obtained from http://scope.aertslab.org/#/Wouters_Human_Melanoma. Patient tumors pre and post MAPK pathway inhibitor therapy, 6 SOX10 knockdown and 6 parental melanoma cell lines, 53 human melanoma cell lines, and 4 mouse melanoma cell lines data are available in the SRA database under accession numbers SRP029434, SRP215051, SRP074198, and SRP247646, respectively. Human melanoma cell line annotation data were gathered from https://ars.els-cdn.com/content/image/1-s2.0-S1535610818301223-mmc2.xlsx. The reporting summary for this Article is available in the Supplementary Information file. All the other data supporting this study are available within this Article, Supplementary Information, Source Data file, or from the corresponding authors upon reasonable requests. Source data are provided with this paper.

## Code availability

Computational analyses were done using publicly available software and R packages or SAS 9.4 (SAS Inc., Cary, NC)

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

## Acknowledgements

This work was supported by grants from American Cancer Society (130042-IRG-16-244-10-IRG), Melanoma Research Foundation and Legacy of Hope Merit Award to C.C. M.H. and A.E.A. were both supported by P01 CA114046-11A1. Additional support was provided by NIH/NCI R01 (CA196278), Department of Defense (CA171056), and Dr. Miriam and Sheldon G. Adelson Medical Research Foundation awards to A.E.A. Additionally, the Melanoma Research Alliance Award #568992 to Drs Melissa Wilson and Andrew Aplin. The Sidney Kimmel Cancer Center Flow Cytometry, Meta-Omics, Translational Pathology, Laboratory Animal and Bio-Imaging core facilities are supported by National Cancer Center Support Grant (P30 CA056036). Support for the Molecular Screening Facility at The Wistar Institute was provided by Cancer Center Support Grant, CA010815. Additional support was provided by NIH/NCI (R01 CA232256) to E.C. The Fox Chase Cancer Center Microscopy/Imaging, Immune Monitoring, Cell Culturing, and Histochemistry facilities used in this study are supported by the Comprehensive Cancer Center Grant NCI (P30 CA06927) and NIH/NCI grant (S10ODO23666). We thank the National Cancer Institute ("NCI") Division of Cancer Treatments and Diagnosis (DCTD) for providing birinapant for in vivo studies. We are grateful to Dr. Barbara Bedogni (University of Miami) and Dr. David Solit (Memorial Sloan-Kettering Cancer Center) for generously providing cell lines. We are grateful to Joel Cassel (The Wistar Institute) for performing the drug screen using an anti-cancer library compound panel and to Dr. Gideon Bollag (Plexxikon Inc., Berkeley CA) for providing PLX4720, PLX2695/PD'901, and PLX8394. We thank Dr. Edward Hartsough (Drexel University) for generating the 1205Lu and PBRTs samples for RNA-seq. At Thomas Jefferson University, we thank Dr. Timothy L. Manser and Trevor Baybutt for the NSG mice colonies, Dr. Paolo Fortina and Dr. Adam Ertel for their help with computational analysis, Dr. Maria Yolanda Covarrubias for her help with confocal microscopy, Dr. Zhijiu Zhong and Raymond O'Neill for helping with immune-histochemistry, and Dr. Lei Yu and Amir Yarmahmoodi for cell sorting. We also acknowledge Dr. Ed Hartsough for generating the PBRT cell lines and sending samples out for RNA-seq.

## Author contributions

C.C. conceived of the study, designed research, performed most of the experiments, analyzed the data, and wrote the manuscript. T.J.P. performed most of the computational analyses present in the manuscript. M.G., S.C., M.T., N.W., D.P., S.R., M.Q.N., and W.C. provided technical support and performed some of the experiments, J.F.B. acquired images with SGH microscope and analyzed the results and performed ECM fibers orientation analysis, R.Z. collect xenograft tumors, G.K. performed computational analysis of one RNA-seq dataset, I.C. and A.S. performed statistical analysis, V.W.R. generated the TMA, A.E.S. helped conceive some of the in vivo experiments, K.H. analyzed IHC slides, X.X. analyzed IHC slides, determined H-scores for TMA and quantified p21 and Ki67 and SOX10 in xenograft tumors slides, E.C. supervised ECM analysis, M.H. provided TMA, A.E.A. conceived of the study, designed research, and wrote the manuscript.

## Competing interests
