## [Peer Review File · Nature Communications]

Reviewers' Comments:

Reviewer #1:

Remarks to the Author:

In this study, Capparelli et al identified SOX10 negative subpopulations in human melanomas. SOX10 knockout in specific human melanoma cell lines resulted in the increased invasiveness, dormancy, resistance to MEKi. SOX10 expression is also downregulated in resistant melanoma cells selected by BRAFi and MEKi treatment. Finally, the authors identified that BIRC3 is upregulated in BRAFi/MEKi resistant, SOX10-low cells, and BIRC3 inhibitor enhanced the efficacy of BRAFi/MEKi in vivo.

This is a sound research with step-by-step rationale, and the identification of BIRC3 inhibitor already used clinically is encouraging. My only major question is the range of indication of this therapeutic discovery. Tsoi et al (Cancer Cell 2018, 33:890–904) and Perez-Guijarro et al (Nature Med 2020, 26:781–791) have identified four melanoma subtypes according to the differentiation status. Among them, the “undifferentiated” subtype has distinctively lowest-to-none SOX10 expression and is resistant to targeted therapies and immune checkpoint blockades. The expression of BIRC3 in the “undifferentiated” subtype, and how such melanoma might respond to BIRC3 inhibitor should be discussed.

Minor comments:

1. Fig. 6G: Please mark the time points of dosing in the figure.
2. Since SOX10-KO cells cannot grow in vivo (Fig. 4B), did tumors relapsed from BRAFi+MEKi and BRAFi+MEKi+Birinapant treatment in Fig. 6G (blue and green curves, respectively) restore SOX10 and/or increase BIRC3 expression?

Reviewer #2:

Remarks to the Author:

This is an elegant piece of work that uncovers a mechanism that ties SOX10 loss, with slow cycling, invasive properties, upregulation of Extracellular matrix components and resistance to therapy. cIAP1/2 inhibitors were uncovered through a small-molecule screen to cause cell death in SOX10 knockout cells and were shown to work synergistically in combination with BRAFi and MEKi to reduce tumor volume.

Although this is not the first study to document the synergistic effect of cIAP1/2 inhibitors with BRAF inhibitors. This has been previously described in colorectal cancer (citation #1). Moreover, as cited by the authors, SOX10 loss has already been implicated in resistance to BRAFi and MEKi (citation #2). However, this is the first study to document and interlink the following characteristics: SOX10 heterogeneity in tumors, and the impact of SOX10 loss on extracellular matrix and invasive properties and minimal residual disease. Then tying these phenotypes to a drug resistant SOX10-deficient subpopulation and linking the synergistic effect of cIAP1/2 inhibitors with BRAFi and MEKi with the ability of cIAP1/2 inhibitors to target a SOX10-deficient subpopulation. In summary, this is a comprehensive study that provides new biological insight into clinically relevant findings in the field. I strongly recommend this study for publication following some minor edits described next:

- The Western blots in Figure 2A include ErbB3 and PDGFR, however these proteins are not discussed in the text until later.
- Figure 3B: Is there a quantification, such as directionality that could be used to complement the pictures of staining to show the “extracellular deposition becomes increasingly more organized”. While FN1 phenotype is obvious, I do not agree with the Collagen IV from the images included. Better images and a quantification of imaging from multiple replicate experiments, would make this finding more sound.
- Figure 6a: SOX10 knockout screen- I could not find any specific methods for this.
- Figure legend 2B: Define NES and BHFDR. i.e. “A heatmap showing GSEA normalized enrichment

scores (NES)...”

- Figure 4D: Results from how many individual tumors (biological replicates), representative images of how many. Images in how many tumors? Quantification doesn't define if values taken from how many different tumors (is each dot a biological or a technical replicate?). The statistical test is not defined, the test should be a nonparametric test as the data looks bimodally distributed.
- Provide a citation: "We note that both of the SOX10-negative melanomas from the single cell sequencing dataset were ICI-resistant"
- Supplementary Fig. 5SE should be referenced earlier -at the start of the section (Sox10 negative cells are sensitive to cIAP inhibitors)- as proof that the inhibitors are in fact working in the system
- Figure 2 legend: Define the control cell line (parental?).
- Figure 4D: methods state that, "areas of interest were selected under a pathologists supervision". Please describe any randomization/blind scoring undertaken to ensure no bias in selecting "areas of interest".
- For CRISPR knockout line generation: individual clones were expanded and tested for knockout. While multiple clones were used, it is well known that clones can have dramatically different growth characteristics. Was a pooled population of CRISPR knockouts ever used for some of these studies. Addition of any data using CRISPR KO pools into the supplementary would be make a convincing addition to the study.
- Does Birinapant affect the ECM remodeling phenotype of SOX10 deficient cells? Staining/microscopy of Birinapant-treated tumors would tie the ECM phenotypes of the first half of the paper with the Sox10-deficient resistant population.
- Likewise, staining for SOX10 status in tumors from Figure 6G would be interesting and may provide extra proof Sox10 deficient cells are being targeted by birinapant/ mechanisms of relapse in the triple-treated tumors.
- Figure 3C, 5A, 6B, 6F are means of at least 3 independent experiments and should all have error bars or at least a version with error bars in the supplementary.
- What are the biological replicates for the in vivo studies?

1. Perimenis, Philippos et al. "IAP antagonists Birinapant and AT-406 efficiently synergise with either TRAIL, BRAF, or BCL-2 inhibitors to sensitise BRAFV600E colorectal tumour cells to apoptosis." *BMC cancer* vol. 16 624. 12 Aug. 2016, doi:10.1186/s12885-016-2606-5

2. Sun C, Wang L, Huang S, Heynen GJ, Prahallad A, Robert C, Haanen J, Blank C, Wesseling J, Willems SM, Zecchin D, Hobor S, Bajpe PK, Lieftink C, Mateus C, Vagner S, Grenrum W, Hofland I, Schlicker A, Wessels LF, Beijersbergen RL, Bardelli A, Di Nicolantonio F, Eggermont AM, Bernards R. Reversible and adaptive resistance to BRAF(V600E) inhibition in melanoma. *Nature*. 2014 Apr 3;508(7494):118-22. doi: 10.1038/nature13121. Epub 2014 Mar 26. PMID: 24670642.

Reviewer #3:

Remarks to the Author:

In this manuscript Capparelli et al. study the effect of disrupting SOX10 in melanoma cells. They present evidence that SOX10 is heterogeneously expressed in human tumors, and that depletion is associated with invasion and quiescence both at the transcriptional and proteomic levels. They then study drug-treated tumors, where they identify a vulnerability of SOX10 depleted cells to cIAP inhibitors. Overall, the breadth of the results is impressive as it spans many methodologies. The manuscript is well-written and clear. We do however have the following concerns:

1. In Fig 2B, which specific clones were used to do the RNA-Seq. Are 2 and 4 the combination of all clones?
2. Clone 4.8 has high expression of SOX10; should it be excluded from the paper since it is a failed clone?
3. The Graf et al paper (ref 14) contradicts the evidence for increase of invasive properties of SOX10 deficient cells. This manuscript provides evidence for SOX10 deficiency promoting invasiveness; while the cited paper shows that SOX10 deficiency reduces invasiveness (<https://pubmed.ncbi.nlm.nih.gov/24608986/>).

4. The authors do not use a non-target gRNA throughout the manuscript as a control for the CRISPR KO of SOX10.
5. Fig 2A should be run on one gel (instead of adding vertical lines). Also it seems strange that Fig 2A and 3A show the same samples blotted for many of the same proteins.
6. In Fig 3B, the immunofluorescence for fibronectin and collagen could be quantified to highlight the difference in mesenchymal state and organization of tumors?
7. In Figure 4C, IHC should also be done for SOX10 to confirm the lack of expression of SOX10 and distinguish the ability of SOX10 deficient cells to form tumors or grow as tumors.
8. In Fig. 5A, 6B replicates should be shown with error bars or show the three individual replicates in the supplement.
9. In the Fig. 5C and 5D experiments, the authors should clarify how these parental lines differ from these resistant tumors? At what time point are these treated tumors collected as their control? It would be more appropriate to use vehicle-treated tumor as a control. Overall, we do not understand the logic of showing the cell lines with and without the MEKi+BRAFi, given that SOX10 is downregulated even without the drugs.
10. In the Figure 5 analysis, are these tumors indeed continuing to grow when treated with drug (progressive disease) or do these resistant tumors neither grow nor shrink (stable disease)? If the former, it seems rather strange for a growing tumor to have depletion of a proliferative signature. How do the authors interpret this?
11. In Fig. 2B and even more so in Figure 5E, the analysis of the gene expression data is far removed from the actual gene expression measurements upon which they are based. In the very least, the expression levels at the pathway level should be shown for both parental and KO or drug-treated, instead of just the ratios. It would be even more convincing to show expression of the individual genes that compose each pathway, especially for figure 2F.
12. In Fig 2F, the caption notes that the top 10 upregulated and downregulated genes are colored in red, but there appears to be more than 10 red dots colored for each group.
13. It would be helpful if the authors showed a dose response curve for A375 during which they derive the resistant tumors from Fig 5C?
14. The authors write: "These data suggest that many of the pathway alterations observed in the CRT lines may be driven by the repression of SOX10." However, the tumors without BRAFi/MEKi do not express SOX10 at baseline (Figure 5C). How do the authors reconcile the lack of SOX10 expression in those untreated tumors and the above referenced statement?

Response to the Reviewers

We thank the reviewers for their careful critique of our manuscript. We have addressed their concerns (see points below) and believe that in doing so we have substantially strengthened the study.

Reviewer #1

Major Comment:

Tsoi et al and Perez-Guijarro et al have identified the “undifferentiated” subtype has distinctively lowest-to-none SOX10 expression and is resistant to targeted therapies and immune checkpoint blockades. The expression of BIRC3 in the “undifferentiated” subtype, and how such melanoma might respond to BIRC3 inhibitor should be discussed.

We thank the reviewer for this very relevant suggestion. Based on this comment, we have included in the revised manuscript RNA-seq data from Tsoi et al. ¹ and Perez-Guijarro et al. ², analyzing the expression of BIRC3 in the different cell states identified in the two manuscripts. The results confirmed upregulation of BIRC3 in the undifferentiated state as well as in the Neural Crest Stem Cell-like (NCSC) state (see page 12).

SOX10 expression is low in the undifferentiated state but maintained in the NCSC state. Since both states are characterized by low levels of MITF, increased BIRC3 expression may be triggered by MITF downregulation. Importantly, both the undifferentiated and NCSC states have been associated with resistance to immune checkpoint blockade and MAPK inhibitors ^{2,3}, mirroring the cross resistance to both therapies observed in patients. These data are now available in Supplementary Figure 5I & 5J and we have added text to the discussion (see page 15-16) to extend our thoughts on the use of cIAP2 inhibitors to prevent/delay resistance to immune-based and kinase inhibitors.

Minor comments:

1. *Fig. 6G: Please mark the time points of dosing in the figure.*

We now clarify in the 6G figure legend that day 0 corresponds to the first day of treatment (see page 44).

2. *Did tumors relapsed from BRAFi+MEKi and BRAFi+MEKi+Birinapant treatment in Fig. 6G restore SOX10 and/or increase BIRC3 expression?*

We appreciate this point and in the revised manuscript we provide IHC data for SOX10 for the queried tumors. SOX10 expression was detected in resistant tumors and we observed a significant increase in SOX10 expression in tumors treated with BRAFi/MEKi/birinapant compared to control tumors (see page 13). These data are now available in Supplementary Figure 5T.

With regards to cIAP2/BIRC3 IHC, unfortunately, upon consultation with Dr. Peter McCue, a pathologist at Thomas Jefferson University, we were not able to obtain specific staining. We used the AF8171 BIRC3 antibody from R&D systems. We decided to not include these data but provide

images of the staining, for the reviewers' reference.

Figure 1. IHC for cIAP2/BIRC3 in A375 xenografts treated as indicated in the figure. Tumors were collected at the end of the experiment as shown in figure 6D within the manuscript.

Reviewer #2

1. *The Western blots in Figure 2A include ErbB3 and PDGFR, however these proteins are not discussed in the text until later.*

We agree with the reviewer and, based on this comment, we removed the ErbB3 and PDGFR β blots from Figure 2A.

2. *Figure 3B: Is there a quantification, such as directionality that could be used to complement the pictures of staining to show the “extracellular deposition becomes increasingly more organized”. While FN1 phenotype is obvious, I do not agree with the Collagen IV from the images included.*

We appreciate this point and point we now include FN1 distribution analysis and quantification in Supplementary Fig. 2C & 2D (see pages 22-23 for Methods). Additionally, we clarified that extracellular deposition of fibronectin appeared increasingly organized in fibrillar structures in SOX10 knockout MeWo cells compared to the parental cells, but no change was observed in Collagen IV (see page 7).

3. *Figure 6a: SOX10 knockout screen- I could not find any specific methods for this.*

We apologize for the lack of information and have added a new paragraph to the Material and Methods of the revised manuscript with information about the assay (see page 22).

4. *Figure legend 2B: Define NES and BHFDR.*

Based on this comment, we have updated the figure legend as follow:

“A heatmap showing GSEA normalized enrichment scores (NES) for the hallmark gene sets collection for MeWo and A375 SOX10 knockout cells (guide #2 and #4) compared to parental cells. NES values are displayed for enriched gene sets, using a Benjamini-Hochberg False Discovery Rate (BHFDR) cutoff of 0.05.”

5. *Figure 4D: Results from how many individual tumors. Quantification doesn't define if values taken from how many different tumors). The statistical test is not defined.*

We thank the reviewer for this comment and apologize for the lack of information. We have revised Figure Legend 4D to clarify that the analysis was performed on three independent tumors, generated either from parental or SOX10 knockout MeWo cells. In total, 12 images from parental and 7 images from SOX10 knockout cells tumors were assessed (see pages 39-40). As indicated in the Materials and Methods: “each area was acquired by collecting 2-4 regions of interest (ROI) using identical settings”. Furthermore, we clarify that statistical analysis was performed using the Wilcoxon Two-Sample test which is indeed a nonparametric test.

6. *Provide a citation: “We note that both of the SOX10-negative melanomas from the single cell sequencing dataset were ICI-resistant”*

We have added this reference.

7. *Supplementary Fig. 5SE should be referenced earlier - as proof that the inhibitors are in fact working in the system*

We agree with the reviewer and have moved Supplementary Figure 5E to Supplementary Figure 5A and modified the rest of Supplementary Figure 5 and the corresponding text accordingly.

8. *Figure 2 legend: Define the control cell line (parental?).*

We apologize for the lack of this information, and we have modified the Figure 2A legend as follows:

“The same number of *parental and SOX10 knockout* cells were seeded in six well plates for each cell line.”

9. *Figure 4D: methods state that, “areas of interest were selected under a pathologists supervision”. Please describe any randomization/blind scoring undertaken to ensure no bias in selecting “areas of interest”.*

We appreciated this comment, and we agree with the reviewer. Based on this comment we have modified the text as follows:

“*Regions of interest were selected by a pathologist (Dr. Xu) who had knowledge of the sample conditions but was blinded to the expected result.*” (see page 23).

10. *For CRISPR knockout line generation was a pooled population of CRISPR knockouts ever used for some of these studies.*

Based on this comment, we now provide Western blot data in Supplementary Figure 2A and growth assay data in Supplementary Figure 4B using a pooled population that constituted of all of the SOX10 knockout clones selected from the MeWo cell line (# 2.1, # 2.2, # 2.8 and # 4.11) mixed at an equal ratio. These studies recapitulate the drug tolerant and invasive phenotype seen in individual MeWo SOX10 knockout CRISPR clones (see page 7).

11. *Does Birinapant affect the ECM remodeling phenotype of SOX10 deficient cells?*

The reviewer raises a very interesting point. To address this, we performed a new *in vitro* experiment to directly analyze the effects of birinapant on ECM proteins in SOX10-deficient cells. The data collected suggests that birinapant treatment reduces ECM protein deposition in MeWo SOX10 knockout cells and A375 CRT35 cells (see page 13). However, a limitation is that these effects may be, in part, due to the ability of birinapant to induce cell death in SOX10-deficient cells. The data are now available in Supplementary Figure 5P & 5Q.

12. *Likewise, staining for SOX10 status in tumors from Figure 6G and may provide extra proof Sox10 deficient cells are being targeted by birinapant.*

Please see Minor comment #2 from Reviewer #1.

13. *Figure 3C, 5A, 6B, 6F are means of at least 3 independent experiments and should all have error bars.*

We agree with the reviewer, and we have modified the graphs to include SD. Furthermore, we provide as Supplementary Information the “Source Data” file containing the raw values.

14. *What are the biological replicates for the in vivo studies?*

We are sorry for the lack of this information in figure 4A & 4B. We have revised the figure and the figure legend text to clarify that 10 and 5 mice per cohorts were used in 4A and 4B, respectively.

Reviewer #3

1. *In Fig 2B, which specific clones were used to do the RNA-Seq. Are 2 and 4 the combination of all clones?*

We thank the reviewer for this comment. In the revised manuscript, we have modified figure legend 2B to clarify that the RNA-seq data shown for MeWo gRNA#2 includes combined data collected from clones # 2.1, # 2.2 and # 2.8. For all of the other samples (MeWo gRNA #4, and A375 gRNA#2 and gRNA#4), shown is the mean from three independent replicates generated for each clone (see page 38).

2. *Clone 4.8 has high expression of SOX10; should it be excluded from the paper.*

Based on this comment, we removed MeWo clone #4.8 from figure 2A.

3. The Graf et al paper (ref 14) contradicts the evidence for increase of invasive properties of SOX10 deficient cells.

We agree with the reviewer and in the Introduction we state: "High SOX10 expression positively regulates melanoma cell proliferation, tumor growth and invasion" and use the Graf et al. paper as reference. Furthermore, based on this comment, we have added the following sentence to the discussion:

"Conversely, a previous study suggests that reduction in SOX10 expression down-regulates invasiveness" (see page 14)."

4. The authors do not use a non-target gRNA throughout the manuscript as a control for the CRISPR KO of SOX10.

In our study we performed a transient transfection with Cas9 and gRNA, followed by a clonal selection and expansion to generate our CRISPR SOX10 knockout clones. Non-targeting gRNAs do not recognize any region within the genome; thus, they should not alter any gene. This means that it would be impossible for us to select and identify clones that underwent the same transfection process as our SOX10 KO clones. Furthermore, in some cases non-targeting gRNAs could result in unintended editing activity, limiting their applicability as a negative control.

We understand that using cells that did not receive any manipulation as a negative control is not ideal, however, it has been recognized as a valid negative control and several publications have used a similar approach⁴⁻⁸. Additionally, since melanoma heterogeneity is a key factor in tumor progression, we believe that using an "untreated" population as a negative control is more relevant for our study.

Lastly, we recognize that inter-clonal heterogeneity could raise some concern and so in the revised manuscript, as suggested by Reviewer #2 (point #10), we now include experiments performed with a pool of SOX10 knockout cells (Supplementary Figure 2A & Supplementary Figure 4B).

5. Fig 2A should be run on one gel (instead of adding vertical lines).

We agree with the reviewer and apologize for the confusion. The vertical lines were used to separate samples run on the same gels but not adjacent to one another. As per journal policy, we now provided a file displaying the uncropped blots.

Also it seems strange that Fig 2A and 3A show the same samples blotted for many of the same proteins.

In light of comment #1 from Reviewer #2, we have removed the ErbB3 and PDGFR β blots from Figure 2A.

6. In Fig 3B, the immunofluorescence for fibronectin and collagen could be quantified to highlight the difference in mesenchymal state and organization of tumors?

Please see comment # 2 from Reviewer # 2.

7. In Figure 4C, IHC should also be done for SOX10 to confirm the lack of expression of SOX10.

The reviewer raised an important point, we now provide the requested IHC for SOX10 in Supplementary Figure 3B. The staining confirmed that the tumors generated by SOX10-deficient cells are indeed negative for SOX10 expression (see page 8).

8. In Fig. 5A, 6B replicates should be shown with error bars or show the three individual replicates in the supplement.

Please see comment #13 from Reviewer #2.

9. In the Fig. 5C and 5D experiments, the authors should clarify how these parental lines differ from these resistant tumors? At what time point are these treated tumors collected as their control?

As described in Sanchez et al.⁹, CRT33 was collected after 91 days and CRT34 and CRT35 after 119 days of treatment with BRAFi and MEKi (PLX4720 200 PPM + PD0325901 7PPM). PBRT #15 and PBRT #16 were isolated from mouse tumor at days 45 and 35 after drug treatment (PLX8394), respectively¹⁰. Isolated cells were subsequently grown *in vitro* in the continuous presence of drug (1uM PLX4720 + 35nM PD032901 for CRTs and 0.5uM PLX8394 for PBRTs).

It would be more appropriate to use vehicle-treated tumor as a control.

We agree with the reviewer, and we now provide Western blot analysis of parental cells and cells derived from a vehicle-treated tumor (CTL), in Supplementary Figure 4F. The results show no difference in SOX10 expression in parental A375 cells and cells derived from a vehicle-treated tumor (see page 9).

Overall, we do not understand the logic of showing the cell lines with and without the MEKi+BRAFi, given that SOX10 is downregulated even without the drugs.

In previous manuscript revisions we have been commonly asked to provide data for tolerant/resistant lines in the presence and absence of the inhibitors, to assess whether the effects observed were transiently induced by drug addiction or adaptive resistance. In our drug tolerant/resistant CRT and PBRT lines, we demonstrate that the removal/addition of inhibitors does not change SOX10 expression or the drug resistant/invasive phenotype. To better clarify this point, we have added the follow sentence in the results section (see page 9):

“The expression of SOX10 was not affected by the presence/absence of the inhibitors, suggesting that the loss of SOX10 is mediated by neither acute drug administration¹¹⁻¹³ nor by addiction to the inhibitors that can develop following acquired resistance^{14,15”}

10. In the Figure 5 analysis, are these tumors progressive disease or stable disease? If the former, it seems rather strange for a growing tumor to have depletion of a proliferative signature. How do the authors interpret this?

Gene Set Enrichment Analysis in Figure 5I was performed on a publicly available RNA-seq dataset previously published by Sun et al.¹⁶. The data were collected from patient samples both before treatment and after resistance to MAPKi occurred. However, regardless of the initial response (initial regression or no response to drug), the tumors were collected while progressing.

The invasive and proliferative gene signatures were generated by analyzing the transcriptomes of melanoma cell lines exhibiting different proliferation rates and invasive characteristics. Importantly, cell lines defined as slow cycling/invasive are still capable of proliferating, but just at a relatively lower rate.

Our analysis of the Sun et al. RNA-seq data suggests that resistance to MAPKi is associated with increased invasion and decreased proliferative signatures. We conclude that resistant tumors have a slower proliferation rate compared to their pre-treatment counterparts. Other studies have previously associated an invasive/slow-cycling phenotype with tolerance and resistance to MAPKi¹⁶⁻¹⁹.

11. In Fig. 2B and even more so in Figure 5E, the expression levels at the pathway level should be shown for both parental and KO or drug-treated, instead of just the ratios. It would be even

more convincing to show expression of the individual genes that compose each pathway, especially for figure 2F.

We thank and agree with the reviewer. To address this point, we now include in the revised manuscript multiple supplemental heatmaps:

Supplementary Figure 1B: heatmaps showing genes commonly enriched in all four gSOX10 comparisons for the four gene sets presented in Figure 2C (EMT, Hypoxia, MYC-1 targets, MYC-2 targets).

Supplementary Figure 1F & 1G: heatmaps showing z-score values for genes in the invasive and proliferative gene sets (presented in Figure 2D) that are enriched in all gSOX10 vs parental comparisons from the A375 (left panel) and MeWo (right panel) RNA-seq data.

Supplementary Figure 4H: heatmap showing GSVA scores for gene sets enriched in all CRT and gSOX10 A375 comparisons, corresponding to Fig 5E.

12. In Fig 2F, the caption notes that the top 10 upregulated and downregulated genes are colored in red, but there appears to be more than 10 red dots colored for each group.

We thank the reviewer for raising this point and apologize for the confusion. The top 10 upregulated and downregulated genes colored in red referred to the top 10 genes for each comparison type (i.e., gSOX10 and CRT). We do understand that this might generate confusion, thus, based on this comment, we have replaced the previous figure with a new one and modified figure legend 5F as follows:

“The top 10 up- and down-regulated genes for CRT and gSOX10 are indicated with gene names. Genes that are common between CRT and gSOX10 are labeled in red.”

13. It would be helpful if the authors showed a dose response curve for A375 during which they derive the resistant tumors from Fig 5C?

We want to clarify that once the CRT lines were extracted from the mouse tumors they were constantly kept in the presence of treatment (1 μ M PLX4720 plus 35 nM PD325901). To further address this comment, we provided below, for the reviewer’s reference, a dose-response curve to BRAFi/MEKi for the CRT34 cell line.

Figure 2. CRT34 cells in IncuCyte Live Cell Analysis System were treated with BRAFi (PLX4720) and MEKi (PD’901) at the indicated concentrations. Cell growth was analyzed as percent plate coverage for 14 days. Treatment was renewed 3 times per week. Shown is the mean \pm SD from three independent experiments.

14. The authors write: “These data suggest that many of the pathway alterations observed in the CRT lines may be driven by the repression of SOX10.” However, the tumors without BRAFi/MEKi do not express SOX10 at baseline (Figure 5C). How do the authors reconcile the lack of SOX10 expression in those untreated tumors and the above referenced statement?

The CRTs shown in Figure 5C are tolerant/resistant cell lines generated from *in vivo* tumors following chronic exposure to BRAFi/MEKi. The label +/- BRAFi/MEKi in Figure 5C refers to a short-term treatment performed on these SOX10-deficient, tolerant/resistant cell lines. In our experiments, we implant A375 cells as xenografts in mice, treat them with BRAFi/MEKi until they develop drug tolerance/resistance, and subsequently culture the tolerant/resistant cells *in vitro*. These resistant lines do not express SOX10. This phenotype appears to be relatively stable, as SOX10 expression in the CRT lines is not affected by the addition or removal of BRAFi/MEKi, as shown in Figure 5C.

References

- 1 Tsoi, J. *et al.* Multi-stage Differentiation Defines Melanoma Subtypes with Differential Vulnerability to Drug-Induced Iron-Dependent Oxidative Stress. *Cancer Cell* **33**, 890-904 e895, doi:10.1016/j.ccell.2018.03.017 (2018).
- 2 Perez-Guijarro, E. *et al.* Multimodel preclinical platform predicts clinical response of melanoma to immunotherapy. *Nat Med* **26**, 781-791, doi:10.1038/s41591-020-0818-3 (2020).
- 3 Rambow, F. *et al.* Toward Minimal Residual Disease-Directed Therapy in Melanoma. *Cell* **174**, 843-855 e819, doi:10.1016/j.cell.2018.06.025 (2018).
- 4 Mainardi, S. *et al.* SHP2 is required for growth of KRAS-mutant non-small-cell lung cancer in vivo. *Nat Med* **24**, 961-967, doi:10.1038/s41591-018-0023-9 (2018).
- 5 Li, C. W. *et al.* Glycosylation and stabilization of programmed death ligand-1 suppresses T-cell activity. *Nat Commun* **7**, 12632, doi:10.1038/ncomms12632 (2016).
- 6 Wang, W. *et al.* ABL1-dependent OTULIN phosphorylation promotes genotoxic Wnt/beta-catenin activation to enhance drug resistance in breast cancers. *Nat Commun* **11**, 3965, doi:10.1038/s41467-020-17770-9 (2020).
- 7 Feng, D. *et al.* BRAF(V600E)-induced, tumor intrinsic PD-L1 can regulate chemotherapy-induced apoptosis in human colon cancer cells and in tumor xenografts. *Oncogene* **38**, 6752-6766, doi:10.1038/s41388-019-0919-y (2019).
- 8 Sun, L. *et al.* Irreversible JNK blockade overcomes PD-L1-mediated resistance to chemotherapy in colorectal cancer. *Oncogene* **40**, 5105-5115, doi:10.1038/s41388-021-01910-6 (2021).
- 9 Sanchez, I. M. *et al.* In Vivo ERK1/2 Reporter Predictively Models Response and Resistance to Combined BRAF and MEK Inhibitors in Melanoma. *Mol Cancer Ther* **18**, 1637-1648, doi:10.1158/1535-7163.MCT-18-1056 (2019).
- 10 Hartsough, E. J. *et al.* Response and Resistance to Paradox-Breaking BRAF Inhibitor in Melanomas In Vivo and Ex Vivo. *Mol Cancer Ther* **17**, 84-95, doi:10.1158/1535-7163.MCT-17-0705 (2018).
- 11 Capparelli, C. *et al.* ErbB3 Targeting Enhances the Effects of MEK Inhibitor in Wild-Type BRAF/NRAS Melanoma. *Cancer Res* **78**, 5680-5693, doi:10.1158/0008-5472.CAN-18-1001 (2018).
- 12 Abel, E. V. *et al.* Melanoma adapts to RAF/MEK inhibitors through FOXD3-mediated upregulation of ERBB3. *J Clin Invest* **123**, 2155-2168, doi:65780 [pii]10.1172/JCI65780 (2013).
- 13 Lito, P. *et al.* Relief of profound feedback inhibition of mitogenic signaling by RAF inhibitors attenuates their activity in BRAFV600E melanomas. *Cancer Cell* **22**, 668-682, doi:S1535-6108(12)00440-0 [pii]10.1016/j.ccr.2012.10.009 (2012).
- 14 Moriceau, G. *et al.* Tunable-combinatorial mechanisms of acquired resistance limit the efficacy of BRAF/MEK cotargeting but result in melanoma drug addiction. *Cancer Cell* **27**, 240-256, doi:10.1016/j.ccell.2014.11.018 (2015).
- 15 Das Thakur, M. *et al.* Modelling vemurafenib resistance in melanoma reveals a strategy to forestall drug resistance. *Nature* **494**, 251-255, doi:10.1038/nature11814 (2013).

- 16 Sun, C. *et al.* Reversible and adaptive resistance to BRAF(V600E) inhibition in melanoma. *Nature* **508**, 118-122, doi:nature13121 [pii]10.1038/nature13121 (2014).
- 17 Roesch, A. *et al.* Overcoming intrinsic multidrug resistance in melanoma by blocking the mitochondrial respiratory chain of slow-cycling JARID1B(high) cells. *Cancer Cell* **23**, 811-825, doi:10.1016/j.ccr.2013.05.003 (2013).
- 18 Muller, J. *et al.* Low MITF/AXL ratio predicts early resistance to multiple targeted drugs in melanoma. *Nat Commun* **5**, 5712, doi:ncomms6712 [pii]10.1038/ncomms6712.
- 19 Song, C. *et al.* Recurrent Tumor Cell-Intrinsic and -Extrinsic Alterations during MAPKi-Induced Melanoma Regression and Early Adaptation. *Cancer Discov* **7**, 1248-1265, doi:10.1158/2159-8290.CD-17-0401 (2017).

Reviewers' Comments:

Reviewer #1:

Remarks to the Author:

The authors have addressed all the reviewer's comments/questions. I recommend this manuscript to be accepted for publication.

Reviewer #2:

Remarks to the Author:

All my previous concerns have been addressed in the revised manuscript and I believe this manuscript will be a stellar contribution to the Nature Communication journal. I strongly accept this manuscript.

Reviewer #3:

Remarks to the Author:

The have addressed very well my concerns from the previous review. I commend the authors on a professional revision and am happy to recommend publication at this time.

REVIEWERS' COMMENTS

Reviewer #1 (Remarks to the Author):

The authors have addressed all the reviewer's comments/questions. I recommend this manuscript to be accepted for publication.

Reviewer #2 (Remarks to the Author):

All my previous concerns have been addressed in the revised manuscript and I believe this manuscript will be a stellar contribution to the Nature Communication journal. I strongly accept this manuscript.

Reviewer #3 (Remarks to the Author):

The have addressed very well my concerns from the previous review. I commend the authors on a professional revision and am happy to recommend publication at this time.

Response to the Reviewers

We thank the reviewers for the kind comments.